# The evolution of trait variance creates a tension between species diversity and functional diversity

György Barabás [1,2 ✉], Christine Parent [3,4], Andrew Kraemer [5], Frederik Van de Perre [6] & Frederik De Laender [7,8,9]

It seems intuitively obvious that species diversity promotes functional diversity: communities with more plant species imply more varied plant leaf chemistry, more species of crops provide more kinds of food, etc. Recent literature has nuanced this view, showing how the relationship between the two can be modulated along latitudinal or environmental gradients. Here we show that even without such effects, the evolution of functional trait variance can erase or even reverse the expected positive relationship between species- and functional diversity. We present theory showing that trait-based eco-evolutionary processes force species to evolve narrower trait breadths in more tightly packed, species-rich communities, in their effort to avoid competition with neighboring species. This effect is so strong that it leads to an overall reduction in trait space coverage whenever a new species establishes. Empirical data from land snail communities on the Galápagos Islands are consistent with this claim. The finding that the relationship between species- and functional diversity can be negative implies that trait data from species-poor communities may misjudge functional diversity in species-rich ones, and vice versa.

[1] Division of Theoretical Biology, Dept. IFM, Linköping University, SE-58183 Linköping, Sweden. [2] ELTE-MTA Theoretical Biology and Evolutionary Ecology Research Group, Budapest, Hungary. [3] Department of Biological Sciences, University of Idaho, 875 Perimeter Dr, Moscow, ID 83844, USA. [4] Institute for Interdisciplinary Data Sciences, University of Idaho, 875 Perimeter Dr, Moscow, ID 83844, USA. [5] Department of Biology, Creighton University, 2500 California Plaza, Omaha, NE 68178, USA. [6] Evolutionary Ecology Group, Universiteit Antwerpen, Universiteitsplein 1, 2610 Wilrijk, Belgium. [7] Research Unit of Environmental and Evolutionary Biology (URBE), University of Namur, Namur, Belgium. [8] Institute of Complex Systems (naXys), University of Namur, Namur, Belgium. [9] Institute of Life, Earth and the Environment (ILEE), University of Namur, Namur, Belgium. ✉email: gyorgy.barabas@liu.se

Functional traits are organismal traits impacting the ecological performance (fitness) of individuals[1,2]. They determine the way individuals interact within the community[3], and contribute to ecosystem functioning[4] and services[5]. The extent to which the available functional trait space is covered by the community is the community's functional diversity[6–9], which is an important indicator of community structure and ecosystem health[8,10] (see Supplementary Note 5 for various diversity measures).

Consequently, functional diversity has been used as a surrogate variable to understand how species loss causes loss of ecosystem function. In-plant ecology, for example, a variety of traits have been described in detail[11] and used to measure how functional diversity changes along environmental gradients brought about by drivers such as elevation and habitat or climate change[12–14]. Likewise, changes in functional diversity signal ecosystem consequences of manual species removal in controlled biodiversity-ecosystem function studies[15]. The last two decades have seen an expansion of functional diversity indices to encompass trait variation among individuals, arguing that species means alone do not suffice to predict assembly, coexistence, biodiversity, and ecosystem functions[16–18].

Because species diversity is often easier to measure, its use as a surrogate for functional diversity has often been examined and debated. However, the covariance between the two kinds of diversity along environmental gradients depends on the specifics of the ecological and environmental scenario, making their relationship nontrivial[13]. Nevertheless, in controlled biodiversity experiments, where one manually creates species diversity gradients, species diversity remains a prime indicator of functional diversity. All else being equal (i.e., without any covarying environmental variables), a greater species diversity often implies greater functional diversity[19], even though the exact nature of this positive relationship might depend on the prevailing environmental conditions[20].

The positive association between species diversity and functional diversity in controlled diversity experiments underpins key studies in ecosystem and community ecology. For example, ecosystems with greater species richness or evenness produce more biomass than monocultures, and do so more stably, because of a greater diversity of resource use strategies[15,21,22]. However, as we argue here, when also considering evolutionary processes the association may be more nuanced. This is because the same intraspecific trait variation that shapes functional diversity is also being affected by species diversity[18,23,24]. Greater species diversity may imply that individuals must avoid overly strong interactions with individuals from other species in order to persist. This mechanism would lead to narrower intraspecific trait breadths. Indeed, a negative relationship between species diversity and intraspecific trait variance has been observed empirically[24]. What remains unclear is the consequence of this effect on functional diversity at the community level. While trait narrowing can be due to plasticity, for example via behavioral changes or changes of resource preference[25,26], it can in principle also occur via evolution acting on heritable phenotypic variation. We intuit that a sufficiently strong decrease of intraspecific trait variation can disrupt the positive effect of species diversity on functional diversity, leading to no relationship at all or even a negative one. However, whether this intuition is theoretically or empirically supported is unknown.

How species diversity influences functional diversity on evolutionary timescales is elusive, because there is no theory on how intraspecific variation of multiple traits evolves against a backdrop of species interactions. From an empirical standpoint, the suite of local and regional processes driving the two kinds of diversity in natural communities[13], as well as landscape connectivity, hamper any direct observation of the relationship between them. One possible approach is to identify a controlled biodiversity experiment lasting long enough for evolution to act upon trait variance. Such an experiment would consist of a set of communities varying in species diversity but otherwise having (near-)identical conditions. Critically, these communities need to be spatially segregated so as to ensure a sufficiently long time without gene flow among the communities, which would blur putative relationships between evolved functional diversity and species diversity. Then, if individuals vary in some functional traits that also mediate competition, and these traits have been recorded at the individual level, we can meaningfully evaluate how species- and functional diversity are related across communities.

Here we present theory and empirical data to test if greater species diversity increases functional diversity on evolutionary timescales. We first present a model that tracks intraspecific variation for an arbitrary number of traits, taking into account species interactions. The model is formulated in continuous time, which permits efficient exploration of how both diversities relate across broad parameter ranges. Next, we present data from a unique natural evolution experiment that conforms to the assumptions of our model in the way outlined above, and for which the necessary data are available: the endemic land snails of the Galápagos Islands, from the genus *Naesiotus* (Supplementary Note 7). Both theory and data support our conclusion that species diversity does not beget functional diversity.

## Results

**Model description.** Our eco-evolutionary model integrates the evolution of an arbitrary number of traits[27,28] as well as species interactions[29,30] in continuous time. The model is based on quantitative genetics: it tracks (i) the population density, (ii) the mean, and (iii) variance (or covariance matrix, in multiple trait dimensions) of each species' trait distribution, assumed to be normal. These three quantities for each species fully characterize the state of the community (i.e., how individuals are distributed across trait space). Trait evolution and density change are driven by phenotypes experiencing differential intrinsic growth depending on their position in trait space and by competition arising through the consumption of shared resources (Supplementary Notes 2–4).

We begin by presenting a motivating example output of this model. We compare two communities, one with three and another with six extant species (Fig. 1A, B). Although the three-species community has lower species diversity, it is clearly more functionally diverse, as it covers a larger proportion of the trait axis. The reason is that its species have evolved much broader trait distributions. A similar example is shown in Fig. 1C, D, in a two-dimensional trait space.

These examples are representative outcomes of our model, seeded with different initial numbers of species and returning communities that have reached an ecological and evolutionary equilibrium. The reason communities with more species diversity turn out less functionally diverse (Fig. 1) is that they are more tightly packed with species. In such communities, species avoid competition by evolving very narrow trait variances, thus reducing their overlap with neighboring species. The stark reduction in species' trait breadths can be understood by visualizing the fitness function over phenotypes (Fig. 2). When species are tightly packed (Fig. 2B), competition is so strong everywhere that the fitness function is always negative except at species' mean trait values, where it is zero. This means that every trait value of every species is selected against, except for their means. In time, this negative selection reduces the genetic

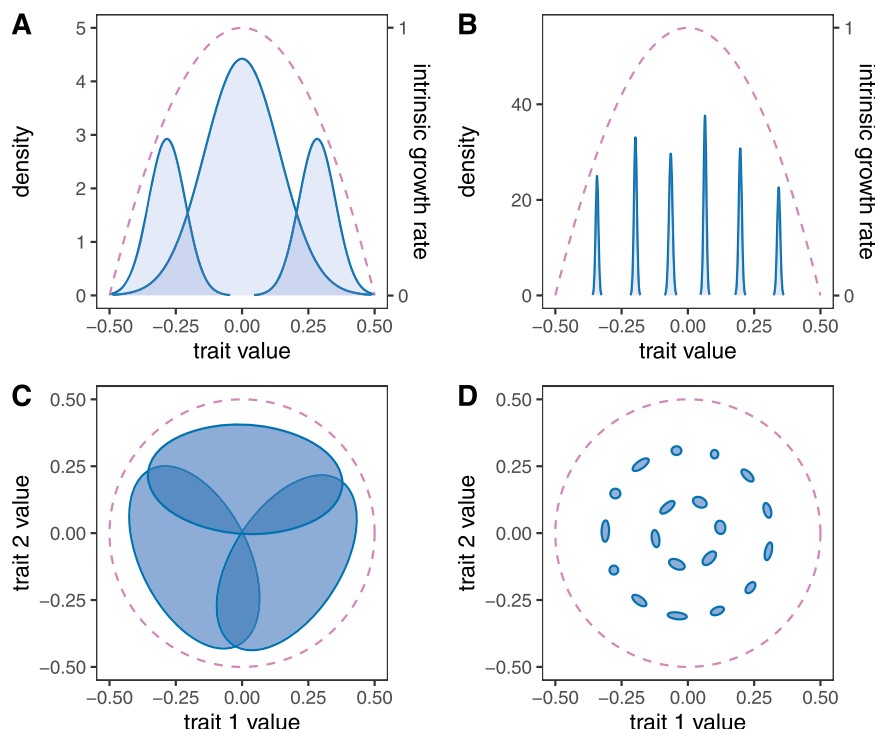

**Fig. 1 Greater species diversity does not necessarily translate to greater functional trait diversity.** Panels show equilibrium states of simulated communities in one (**A**, **B**) and two (**C**, **D**) trait dimensions, with low (**A**, **C**) and high (**B**, **D**) final species diversity. In **A** and **B**, density is plotted along the ordinate and trait value along the abscissa. The blue shaded curves are the trait distributions of different species; the area under each curve is the total population density of the corresponding species. The dashed lines show what the rate of exponential growth of the given phenotype would be in the absence of competition (right-hand ordinate). The competition width (trait distance beyond which competition is significantly reduced between two phenotypes; Supplementary Notes 3 and 6.1) is 0.15. **C**, **D** The axes correspond to the two traits and the ellipses represent the trait distribution contours of different species; the contour lines are the 95% regions of the trait covariance. The dashed line encloses the region of trait space where intrinsic growth rates are positive; the competition width is 0.2.

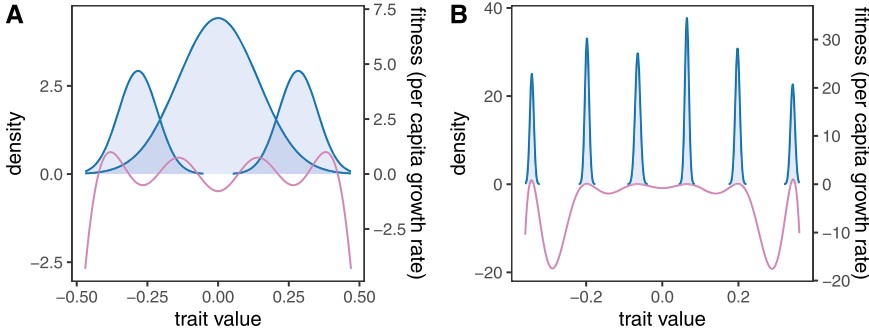

**Fig. 2 Graphical overview of why species evolve narrower trait breadths in more tightly packed communities.** Shaded curves are species' equilibrium trait distributions as in Fig. 1; the purple lines are the per capita growth rate of the corresponding phenotype, independent of species identity. This fitness curve (right-hand ordinate) is obtained from Eq. (4) in the Methods. **A** With fewer species, there are regions with both positive and negative growth along the trait axis, which will act to increase and decrease trait density in those regions, respectively. Since the trait distributions are forced to be Gaussian by random mating, this will have the effect of increasing the trait variance to the point where this increase is counteracted by negative selection acting on extreme trait values. **B** In tightly packed communities, each phenotype experiences negative growth except at species' mean trait values where growth is zero. This means that selection eventually removes all genetic variance. Any remaining trait variation is environmental; i.e., stemming from developmental noise and other sources unaffected by selection.

variance of each species to zero, leaving only the environmental component of trait variation (Supplementary Note 1).

By contrast, for low species diversity (Fig. 2A), the trait space has regions of both negative growths (in densely populated regions) and positive growth (in areas with a lower prevalence of individuals, farther away from the densely occupied regions). The equilibrium trait breadth of each species then emerges from phenotypes with positive growth becoming more abundant, and

phenotypes with negative growth becoming less abundant, and the distribution as a whole is kept in balance by the constraint that the overall intraspecific trait distribution must be Gaussian (a consequence of the quantitative genetic assumptions and random mating). Consider the leftmost species in Fig. 2A as an example. Within this species, individuals with trait values close to the species-level mean around −0.25 are selected against (have negative growth rate), and individuals with either slightly larger

or smaller trait values are selected for (local maxima of the purple fitness curve). Individuals that differ even more from the species mean are again selected against. Intuitively, one would expect this to result in the trait distribution becoming bimodal with time. But it does not, due to random mating constantly restoring the normal shape of the trait distribution. Instead, the trait variance increases. Yet, this increase does not continue indefinitely because individuals with sufficiently extreme trait values have negative fitness: for very large trait variances, the negative selection against extreme trait values will reduce trait variance. The balance of negative selection and random mating then determines the trait variance achieved at equilibrium.

When trait space is multidimensional, the model is agnostic about any correlations between the trait dimensions. This means that, while no genetic correlation structure is imposed by hand, it may still emerge in response to the selective pressure exerted by, e.g., other species in the community. An example is provided in Fig. 1C. There is nothing in the model that would prescribe the species to adopt one genetic covariance structure or another. Yet, the three surviving species each assume a visibly nonzero covariance (zero covariance would correspond to the 95% contour lines being circles instead of ellipses). This emerges because although the species would want to evolve towards the middle where intrinsic growth rates are highest, they are prevented from doing so by the others. The best compromise between achieving high growth rates without experiencing too much competition leads to the emergence of a nonzero genetic covariance structure for each species, in this example.

**Model predictions.** The outcome illustrated in Figs. 1–2 is robust to changes in the model's parameters and assumptions: functional diversity never increases monotonically with, and often declines with, species diversity (Fig. 3). These results emerge from a numerical experimental setup whereby we vary the number of initial species, the number of trait dimensions, the amount of environmental trait covariance per species, the initial genetic covariance levels, the competition width, the shape of the intrinsic growth function, and the diversity metric used (Methods and Supplementary Note 6). Additionally, we compare these results to runs of the same model but with no evolution of trait breadths allowed. As expected, this leads back to a situation where species- and functional diversity are positively related, underlining the fact that it is indeed the evolution of trait covariances that is responsible for the reduction in functional diversity.

In Fig. 3 the effect of species diversity on functional diversity is weaker for larger environmental trait covariances. This stands to reason, as the total phenotypic covariance is the sum of the genetic and environmental covariances (Supplementary Note 1), of which only the former can evolve. The smallest the genetic covariances can evolve to be is zero, in which case the total phenotypic covariance will still be equal to the environmental covariance. The latter is not necessarily small, which is why functional diversity drops less with species diversity in the presence of higher environmental covariances. Thus, if the environmental covariances are large, then no matter how small the genetic components evolve to be, the actual trait covariance (and therefore functional diversity) will still remain substantial.

We also checked the robustness of our results against changes in the model structure. Thus far we have assumed symmetric and localized competition. Symmetric competition means that the effect of one phenotype on another is always the same as vice versa. Competition is localized when two phenotypes only interact to an appreciable degree if they are sufficiently close in trait space: competition decreases with trait distance. Localized competition arises naturally when the strength of interaction

depends on resource overlap between two phenotypes. Nonlocal competition can arise through mechanisms that lead to a competitive hierarchy, such as light competition between trees of different height[31]. Species higher up in the hierarchy competitively affect all lower species, but the weaker competitors do not affect stronger ones much. In return, competitively inferior species often have to compensate for advantages such as better colonization abilities or higher intrinsic growth rates, leading to competition-colonization[32] and competition-mortality[33] tradeoffs, respectively.

To explore their influence on model predictions, we have implemented nonsymmetric and nonlocalized competitive structures in our model (Supplementary Note 6.4). Allowing for asymmetric competition did not generate any qualitative differences from our results (Supplementary Figure 11, top row): there is still an overall negative relationship between species- and functional diversity. However, nonlocal competition via a competition-mortality tradeoff[33] turns the relationship positive (Supplementary Figure 11, bottom row). The trait pattern underlying this difference is illustrated in Supplementary Figure 12: the lower a species lies in the hierarchy (corresponding to larger mean trait values), the broader its trait distribution becomes. This happens regardless of the number of species, and thus weak competitors evolve to occupy a large chunk of trait space even in the presence of high species diversity. Thus, the narrowing of the trait distributions observed under localized competition no longer occurs, leading to an overall positive relationship between species- and functional diversity.

Returning to our basic assumption of localized competition: a potentially negative relationship between species- and functional diversity has implications for the study of ecosystem functioning. However, whether these implications are immediate or obvious depends on how function scales with traits. In case a given ecosystem function is directly proportional to available functional diversity[34], lower functional diversity by definition maps to lower ecosystem function, and therefore increased species diversity (implying lower functional diversity) degrades ecosystem functioning. Otherwise, one must first determine how function depends on available traits. For instance, our model can be used to explore two commonly studied ecosystem functions: resource use and biomass production[15,22]. Even with trait variance evolution reducing functional diversity, species diversity continues to promote these two functions (Fig. 4; Supplementary Note 6.3). The reason is that the low functional diversity found in communities with high species diversity is still sufficiently high to achieve positive complementarity[35]. In other words, we find that ecosystem functioning is invariably enhanced by species diversity, despite the fact that functional trait diversity actually diminishes. Thus, species diversity remains beneficial for ecosystem functioning—but crucially, not because functional diversity increases with species diversity.

**Empirical evidence.** The theorized mechanism regulating functional diversity over evolutionary time should leave its mark on any community conforming to its assumptions. However, directly quantifying its consequence for the effect of species on functional diversity is not straightforward: diversity patterns in natural communities are typically driven by a suite of other local and regional processes as well. One possible approach to control for these effects is to identify a suitable "natural evolution experiment": a set of isolated subcommunities that have evolved independently for a long time, vary in the species diversity of a focal functional group, but otherwise harbor near-identical environmental conditions. Then, if individuals vary in some functional traits which also mediate competition, and these traits have been

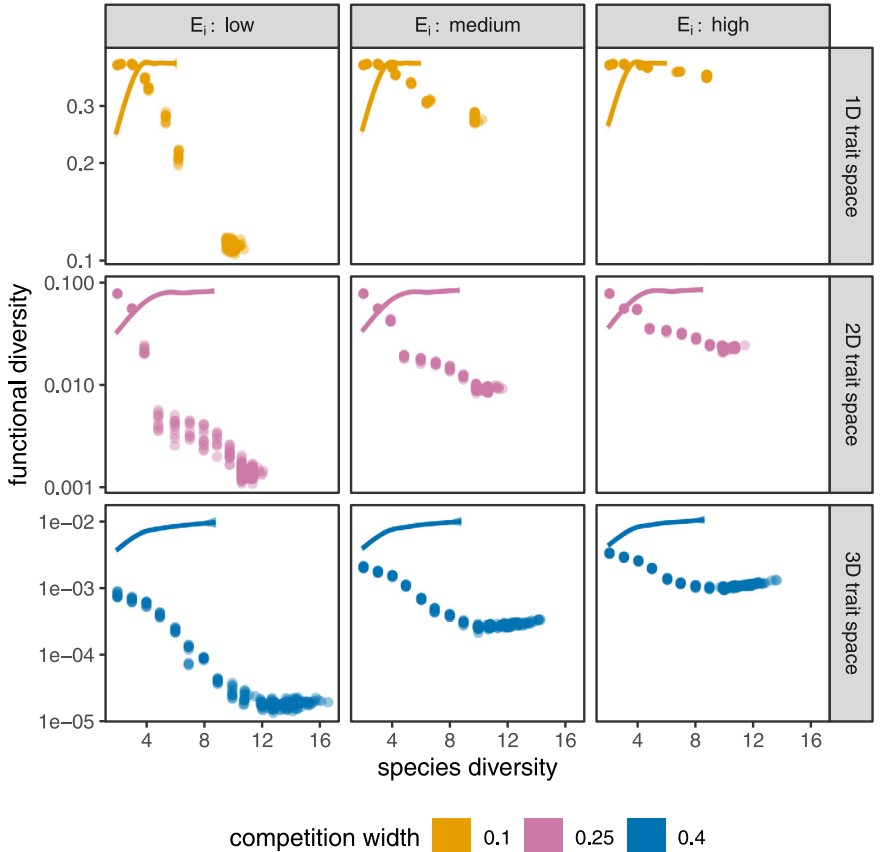

**Fig. 3 Functional diversity plotted against species diversity, for various dimensions of trait space (rows), levels of environmental trait variance (columns), and competition widths (colors).** Each point is a single replicate model run, across 10 replicates per parameterization; $n = 240$ independent samples per panel. Functional diversity was obtained by dividing each trait dimension into 101 equally-sized bins in the $[-1, 1]$ range (Supplementary Note 5; note the log scale along the ordinate). Since higher-dimensional trait spaces have more room and therefore can harbor more species all other things equal, the competition widths were chosen larger for higher-dimensional trait spaces to create comparable species diversities for different trait dimensions. The lines in each panel are locally weighted polynomial regression fits on corresponding results from an appropriate null model with 99% confidence intervals around them (these errors are so small that they are barely visible). The null model consists of an exact re-run of the replicates, with the sole difference that the trait covariances are not allowed to evolve. In the null model, the relationship between species- and functional diversity is positive, as expected. They do not reach as far along the x axis because without the trait breadths being allowed to shrink, only fewer species can be packed into the trait space, limiting species diversity.

recorded at the individual level, we can evaluate whether relatively more species-diverse subcommunities tend to have relatively lower functional diversity.

One natural evolution experiment which conforms to the assumptions of our model and for which the necessary data are available is a dataset of endemic land snails on the Galápagos Islands (Supplementary Note 7), from the genus *Naesiotus*. Previous work has found that intraspecific variation in the land snails' shell morphology is positively correlated with habitat heterogeneity and negatively correlated with the number of co-occurring congener species. These together suggest that snails compete for available niche space, and that shell morphology reflects adaptedness to those ecological opportunities[36]. Displaying the distribution of individuals in each subcommunity reveals that species segregate in the two-dimensional trait space spanned by shell centroid size and shell shape (measured by the first PC axis explaining over 80% of shape variation), further supporting the idea that shell morphology, quantified by these two traits, mediates competition (Supplementary Figure 13). Community age varies from 60 thousand to over three million years across the islands[36,37]. Such long time spans are sufficient for substantial shell morphology evolution to have taken place. Some unrelated species occurring on different islands have highly

similar shell morphotypes[38], strongly implying that evolution has converged on similar solutions, and therefore that the species form evolutionarily stable communities.

There are thirteen islands in the dataset, most of which possess both arid and humid habitat zones. The distribution ranges of snail species never overlap across them. Thus, their species do not have the opportunity to interact, and so the compositions of the humid and arid zones form effectively separate communities. We used the shell morphology of individual snails to compute functional diversity for each subcommunity (i.e., island-habitat zone combination). Shell morphology is a strong indicator of habitat specialization in this system. Snails with long, slender shells live in dry habitats; snails with round shells live in humid habitats; and intermediate varieties live in semi-arid environments[39,40]. There is additionally a tight correlation between shell color and local background color[41]. Snails tend to partition habitat based on structures and surfaces available (e.g., under rocks or logs, low on tree trunks, on small low vegetation) and therefore, although the snails are not host-specialists, they have clear microhabitat preferences. Since various plant species provide distinct microhabitats, ecological opportunities are thought to roughly scale with the number of available host plant species per community[36]. For this reason, to appropriately

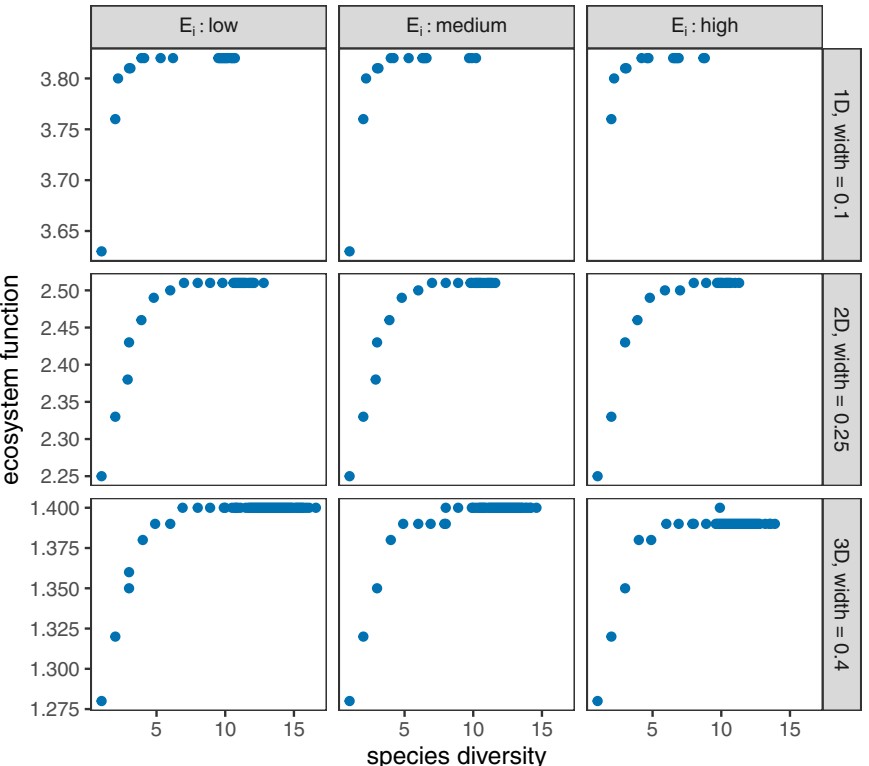

**Fig. 4 Ecosystem functioning against species diversity, for various values of trait space dimensionality and competition width (rows) and levels of environmental trait variation (columns).** Data (points; $n = 480$ independent samples per panel) were calculated from the same simulation results that were used in Fig. 3. Ecosystem functioning is measured either by total resource use or biomass production (Supplementary Note 6.3). It always increases with increasing species diversity—despite the fact that functional diversity itself declines with species diversity, as shown by Fig. 3.

evaluate species diversity, we accounted for the (sometimes vastly) varying habitat heterogeneity across subcommunities by normalizing species diversity with the number of available host plant species.

One limitation of the dataset is that the recorded relative species abundances do not reflect the actual ones on the islands, hampering any diversity calculations. We therefore also tested the sensitivity of the empirical relationship between species- and functional diversity to randomizing these relative abundances.

The data do not support a positive relationship between functional diversity and species diversity per host plant species (Fig. 5). Even if one does not correct for the number of host plant species and considers raw species diversity, no positive relationship is found (Supplementary Note 7.2). Randomizing relative abundances does not change this result, showing that the above limitation of the dataset does not affect our conclusions (Supplementary Note 7.2).

**Discussion**

We presented an eco-evolutionary community model showing that over evolutionary timescales species diversity does not necessarily promote, and may even decrease, functional diversity. This theoretical finding proved robust against alterations of model parameters and structure, as long as competition was localized (only sufficiently similar phenotypes interact with one another). A dataset of shell morphology measured across thirteen independently evolved communities of endemic land snails confirmed that islands with greater snail species diversity do not necessarily harbor greater functional diversity of shell morphology.

The possibility for phenotypic variance to evolve in a community context is what causes the model to often predict weaker

functional diversity at greater species diversity (Supplementary Notes 1–2). The integration of both features—trait variance evolution and the community context—sets apart the present model from existing eco-evolutionary approaches, which either keep this variance constant[30,42,43] or focus on the dynamics of a single species only[27,28]. Our simulations and analyses show that only when both features are simultaneously present do we predict a tension between species- and functional diversity. This observation highlights how the interplay of ecological factors (the community context) and evolutionary ones (trait variance evolution) results in predictions of biological diversity that are qualitatively different from those resulting from considering either factor alone[44].

While model outcomes were highly robust as long as competition was localized, switching to a hierarchical model of competition qualitatively changed our predictions: it flipped the effect of species diversity on functional diversity from negative to positive (Supplementary Note 6.4). Hierarchical competition means that species can be sequentially ordered in a way that species earlier on the list exert a large competitive effect on all those further back, but the reverse effects are weak at best. Community structures leading to such competition are common in nature, e.g., among herbaceous plants[45] and forest trees[31]. Therefore exploring its effect on eco-evolutionary dynamics is warranted given our numerical results. In particular, it would be interesting to see if, given a dataset with isolated and independently evolved subcommunities of species engaging in hierarchical competition, an analog to our Fig. 5 would reveal a positive relationship between species- and functional diversity. More generally, a natural next step in understanding the relationship between species- and functional diversity under a richer set of community structures is to move away from localized competition in favor of various nonlocal structures, and perhaps

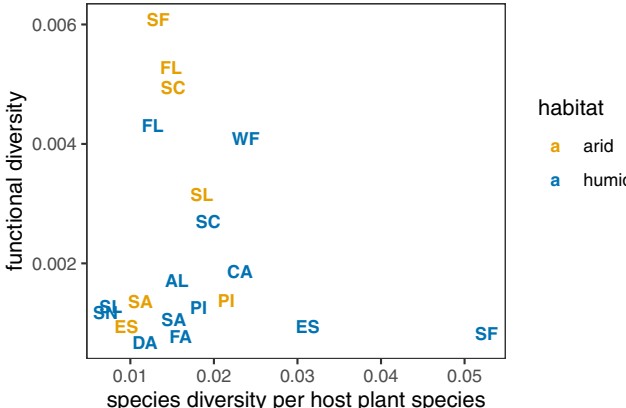

**Fig. 5 Functional diversity plotted against species diversity, for the land snail communities on the Galápagos Islands.** Species diversity is normalized by the number of host plant species in each sub-community, to get a better index of species diversity relative to the number of available ecological opportunities. Labels are island name abbreviations: Alcedo Volcano (AL), Cerro Azul Volcano (CA), Darwin Volcano (DA), Espanola (ES), Fernandina (FA), Floreana (FL), Pinzon (PI), Santiago (SA), Santa Cruz (SC), Santa Fe (SF), San Cristobal (SL), Sierra Negra Volcano (SN), and Wolf Volcano (WF). Colors represent communities in the arid (yellow) and humid (blue) zones of the islands, which form independent subcommunities ($n = 20$ independent samples). The data do not support a positive relationship between species- and functional diversity. (A linear regression has slope $-0.03 \pm 0.04$ with $p = 0.53$; however, since the data are heteroscedastic and the expected patterns from our model as shown in Fig. 3 are not linear, to begin with, any such statistic should be treated as just an illustration.) The lack of a positive trend is retained even if one does not normalize species diversity with the number of host plant species (Supplementary Figure 14). Furthermore, a negative slope is retained in >90% of cases after randomizing the number of sampled individuals (Supplementary Note 7.2). This means that the results are robust against measurement error in relative abundances. [Further statistical information on the regression: $p$ value based on two-sided $t$ test; no correction for multiple comparison].

even non-competitive interactions—as an example, the cascade model in food web theory[46] involves interactions that are non-local and trophic as opposed to competitive.

It has been argued before in the literature that generalist-specialist tradeoffs can facilitate the coexistence of two species via phenotypic subsidy[17]. This means that two species with very similar mean trait values may still coexist if their trait variances are sufficiently distinct, as this allows the more variable species access to resources that the specialist simply cannot utilize. Indeed, this effect was found in an eco-evolutionary model that is similar to ours, except that it assumes fixed, non-evolving trait variances[42]. However, the same study also found that the conditions required for such coexistence to evolve are quite restrictive, especially in multispecies communities. We checked our extensive numerical experiments to see if any communities have evolved where at least one pair of species coexist by virtue of a generalist-specialist tradeoff, but have found none. Quite possibly, the already restrictive conditions for this to emerge with fixed trait variances become even more restrictive when additionally allowing for trait variance evolution. In fact, we have set up two-species simulations with the goal of achieving this type of coexistence but were unable to do so. Obviously, our inability to numerically generate generalist-specialist tradeoffs does not constitute mathematical proof that this is impossible. (Such proof might be difficult to attain: it involves analyzing a six-dimensional dynamical system where the state variables are the population

density, trait mean, and trait variance of each species.) We, therefore, leave the task of either finding such an example or else demonstrating its impossibility as a future challenge.

An important facet of our results is that even with trait variance evolution reducing functional diversity, species diversity continues to promote resource use and biomass production, two commonly studied functions in biodiversity-ecosystem function research[15]. However, our results show that complementarity could have been greater had species not adapted their trait variances to the presence of competitors. In that sense, our results suggest that trait variance evolution weakens biodiversity effects on function over evolutionary timescales. This eco-evolutionary mechanism can both counteract and reinforce strictly ecological mechanisms. For instance, plant-soil feedbacks[47] can intensify plant diversity effects on production with time[48]. Conversely, biodiversity effects on function may weaken with time when competitive exclusion is slow in phytoplankton communities[49], and may even become negative in response to climatic effects such as increased temperatures[50]. Overall, our findings may contribute to a better understanding of how and why biodiversity effects on function change with time[51].

Our model does have some limitations, nuancing the interpretation of the numerical results. First, there is no upper limit to genetic covariances, which is an artifact of having assumed large populations and infinitely many alleles per locus, instead of modeling individuals and their alleles explicitly[52] (Supplementary Note 1). This means that the model may predict increases in trait variance which would not be possible with realistic levels of standing genetic variation. Second, since trait distributions are always normal and therefore unimodal, the model cannot produce speciation (which requires a multimodal trait distribution). That is, in our model, only the parameters of this distribution can evolve (mean and variance), not the type of distribution. Disruptive selection within a single species will therefore increase this species' trait variance without ever splitting into two species. Including speciation into our model, for instance by adopting an adaptive dynamics approach[53,54], would by definition increase species diversity while possibly reducing trait diversity, thus tempering any negative relationship between the two. In sum, the model almost certainly overestimates the reduction in functional diversity, and it is, therefore, safer to say that we expect the lack of a positive relationship, rather than a necessarily negative one, between species- and functional diversity.

The decrease of functional diversity with species richness we find is an unexpected consequence of a well-known phenomenon: the negative effect of interspecific competition on intraspecific trait breadths. There is ample empirical support for this mechanism across a variety of systems. For instance, Bolnick et al.[55] found that intraspecific variation of morphological and behavioral traits expand after experiencing ecological release. More specifically, they showed that this expansion is due to greater between-individual variability, and not to individuals themselves becoming better generalists. That is, a more generalist population achieves its generality by harboring individuals that are more heterogeneous in their resource use, coined the "niche variation hypothesis" by Van Valen[56]. This mechanism is explicitly included in our model, where individuals have one given trait (represented by a single point in the multidimensional trait space), and their diet breadths around that trait are given by the covariance matrix **W** (Methods). A greater population-level diet breadth is thus necessarily created by different individuals having different diets, reflecting Van Valen's hypothesis.

Morphological traits underpin the analysis of the empirical snail data. In the case of the Galápagos land snails, there are multiple lines of evidence that morphological variation effectively correlates with niche variation, which is a prerequisite for useful

analyses of trait-environment interactions[55]. First, the case where individual-environment interactions are driven by non-morphological traits (e.g., behavior) applies less to land snails than to other invertebrates and vertebrates. This is because these snails stay inactive for prolonged periods of time, attached to surfaces for months at a time until it rains. Second, the morphological shell traits we consider (and have been typically considered for this model system) happen to be especially good indicators of the snails' selective environments. Snail shells serve numerous functions: they provide protection against predation[57] and water loss[57,58], and they facilitate heat dissipation[59,60]. These advantages must be balanced against the metabolic costs of building, maintaining, and transporting the shell through a dynamic environment. In particular, shell size and shape are strongly selected by local environmental conditions for many mollusks[61,62]. Finally, the ecological role played by shell morphology is also implied by the cost involved in building elongated shells. Elongated shells are more costly to produce than rounder shells because of their higher surface area to volume ratio[57]. Thus, if building cost was the sole selective pressure, one would predict all snails to have rounder shells. However, elongated shells present important advantages such as decreased water loss and increased heat dissipation, as mentioned above. These are due to their smaller aperture and greater surface-to-volume ratio, respectively[63]. Thus, the very presence of elongated shells implies they must confer an advantage over rounder shapes.

Admittedly, the empirical snail data do not offer direct empirical support for a significant negative relationship between species- and functional diversity, only the lack of a positive one. Yet one can argue that it is a plausible hypothesis that the observed relationship is in fact negative. Three lines of evidence support this conclusion. First, when log-transforming species diversity to create a more homogeneously populated and normally distributed predictor range, there is still no positive relationship, with a linear regression yielding an overall negative slope (Supplementary Figure 16). Second, removing any or both of the two extreme observations (ES humid and SF humid) does not result in a significant positive slope either. In fact, removing only ES humid still gives a negative slope, while removing only SF humid yields a slope of zero. Only when removing both at the same time do we get a positive slope, but the result is non-significant. This suggests that the observed pattern is not being excessively driven by those two points. Third, randomizing species abundances and applying regression analysis (SI, Section 7.2) produced a negative slope in >91% of random trials. That is, the negative slope is very robust to altering the underlying data. While none of these observations force the conclusion that the relationship between species- and functional diversity is negative in the land snail dataset, together they form a compelling case that it is at least a reasonable possibility.

Different ecological processes can lead to different relationships between species- and functional diversity along environmental gradients[13]. Our results show that communities with greater species diversity, but sharing the same environmental conditions, will not necessarily have greater functional diversity. The reason is that evolutionary processes can impede a positive monotonic relationship between species- and functional diversity, and may even turn it into a negative one. This happens via selection for narrower intraspecific trait distributions, curtailing the overall coverage of trait space. This reinforces the message that functional diversity should be measured at the individual level instead of aggregating traits into species averages[17,18,23,24,64]. Crucially, our results highlight another reason why doing so is important: species' contributions to functional diversity obtained in species-poor communities, even when individual-based, can overestimate functional diversity in species-rich communities.

## Methods

**The model**. Our eco-evolutionary model tracks $S$ species in an $L$-dimensional trait space, following their population densities $N_i$, trait mean vectors $\boldsymbol{\mu}_i$ (an $L$-dimensional vector for each species), and genetic covariance matrices $\mathbf{G}_i$ (an $L \times L$ matrix for each species) via the following equations (Supplementary Note 1–2):

$$\frac{dN_i}{dt} = N_i \int r(\mathbf{z}) p_i(\mathbf{z}) \, d\mathbf{z}, \tag{1}$$

$$\frac{d\boldsymbol{\mu}_i}{dt} = \mathbf{G}_i \mathbf{P}_i^{-1} \int (\mathbf{z} - \boldsymbol{\mu}_i) r(\mathbf{z}) p_i(\mathbf{z}) \, d\mathbf{z}, \tag{2}$$

$$\frac{d\mathbf{G}_i}{dt} = \mathbf{G}_i \mathbf{P}_i^{-1} \left[ \frac{1}{2} \int [(\mathbf{z} - \boldsymbol{\mu}_i) \circ (\mathbf{z} - \boldsymbol{\mu}_i) - \mathbf{P}_i] r(\mathbf{z}) p_i(\mathbf{z}) \, d\mathbf{z} \right] \mathbf{P}_i^{-1} \mathbf{G}_i. \tag{3}$$

Here $t$ is time; $\mathbf{P}_i = \mathbf{G}_i + \mathbf{E}_i$ is the total phenotypic covariance matrix of species $i$ where $\mathbf{E}_i$ is the environmental covariance matrix accounting for nonheritable trait variation; $p_i(\mathbf{z})$ is species $i$'s trait distribution function at $\mathbf{z}$, assumed to be multivariate normal with mean $\boldsymbol{\mu}_i$ and covariance matrix $\mathbf{P}_i$; $\circ$ denotes the outer product of two vectors; $r(\mathbf{z})$ is the per capita growth rate of phenotype $\mathbf{z}$ (irrespective of species identity); and the integrals extend along with the whole $L$-dimensional trait space. These growth rates $r(\mathbf{z})$ are derived from a consumer-resource model (Supplementary Note 3), and have the form

$$r(\mathbf{z}) = \int u(\mathbf{z}, \mathbf{y}) R_0(\mathbf{y}) \, d\mathbf{y} - m(\mathbf{z}) - \sum_{j=1}^{S} \iint u(\mathbf{z}, \mathbf{y}) u(\mathbf{z}', \mathbf{y}) N_j p_j(\mathbf{z}') \, d\mathbf{y} \, d\mathbf{z}', \tag{4}$$

where

$$u(\mathbf{z}, \mathbf{y}) = [(2\pi)^L \det(2\mathbf{W})]^{1/4} \mathcal{N}(\mathbf{z}; \mathbf{y}, \mathbf{W}) \tag{5}$$

is phenotype $\mathbf{z}$'s resource utilization function ($\mathcal{N}$ is the multivariate normal distribution with mean $\mathbf{y}$ and covariance $\mathbf{W}$),

$$R_0(\mathbf{y}) = [(2\pi)^L \det(2\mathbf{W})]^{-1/4} \tag{6}$$

is the saturation concentration of resource $\mathbf{y}$, and $m(\mathbf{z})$ is the intrinsic mortality rate of phenotype $\mathbf{z}$. The covariance matrix $\mathbf{W}$ of the utilization function is written as $\mathbf{W} = (\omega^2/4)\mathbf{I}$, where $\mathbf{I}$ is the $L \times L$ identity matrix and $\omega$ the competition width.

**Model simulation protocol**. To explore the model's behavior, the following parameters were varied in a fully factorial combination:

- Number of trait dimensions $L$: either 1, 2, or 3.
- Number of initial species $S$: either 2, 3, …25.
- Initial genetic covariances $\mathbf{G}_i(0)$ are generated via $\mathbf{G}_i(0) = \mathbf{U}_i \mathbf{B}_i \mathbf{U}_i^T$, where $\mathbf{U}_i$ is a random orthogonal matrix and $\mathbf{B}_i$ is diagonal with nonzero entries sampled uniformly and independently from either [0.01, 0.05] (low values), or [0.05, 0.1] (high values).
- Environmental trait covariances $\mathbf{E}_i$: they are diagonal with the diagonal entries sampled uniformly and independently from either [0.005, 0.008] (low), [0.015, 0.018] (medium), or [0.025, 0.028] (high).
- Competition width $\omega$: could take a low or a high value depending on the trait dimensionality $L$:

  for $L = 1$, $\omega$ either 0.1 or 0.15;
  for $L = 2$, $\omega$ either 0.25 or 0.3;
  for $L = 3$, $\omega$ either 0.4 or 0.45.

- Shape of the intrinsic mortality function: either $m(\mathbf{z}) = (\mathbf{z}\mathbf{z})/\theta^2$, or $m(\mathbf{z}) = (\mathbf{z}\mathbf{z})^2/\theta^4$, where $(\mathbf{z}\mathbf{z})$ is the scalar product of $\mathbf{z}$ with itself and $\theta = 1/2$ is a fixed parameter. (All figures in the main text use the former function; see the Supplementary Notes for results with the latter).

This leads to $3 \times 24 \times 2 \times 3 \times 2 \times 2 = 1728$ unique parameter combinations. Ten replicates were run for each parameterization, with different initial trait means (uniformly sampled from $[-0.5, 0.5]$ along each trait dimension for each species). Initial population densities were set to $N_i = 1$. Each replicate was integrated for $10^{10}$ time units, to make sure the model communities reached an eco-evolutionarily stable equilibrium.

**Measuring species- and functional diversity**. Species diversity was obtained via Hill numbers of order $q$:

$$^qD = \left( \sum_{i=1}^{S} f_i^q \right)^{\frac{1}{1-q}}. \tag{7}$$

This formula returns the usual inverse Simpson index for $q = 2$, and the exponential of the Shannon index in the limit of $q \to 1$. For functional diversity, we use metrics estimating the fraction of the total trait space covered, as well as the evenness of this cover (Supplementary Note 5). To do so, we first obtain the community-wide trait probability density function $\mathcal{D}(\mathbf{z}) = \sum_{i=1}^{S} f_i p_i(\mathbf{z})$ using species' trait distributions $p_i(\mathbf{z})$ and relative frequencies $f_i = N_i / \sum_{j=1}^{S} N_j$. We then divide the $L$-dimensional trait space into a grid of $C$ small (hyper-)cubes, evaluate

$\mathcal{D}(\mathbf{z})$ at the center of each, and normalize the result by the sum of all values to obtain a relative density $\hat{\mathcal{D}}_i$ in each cubic cell $i$. We then obtain the community's functional diversity of order $q$ as

$$^q D = \lim_{C \to \infty} \frac{1}{C} \left( \sum_{i=1}^{C} \hat{\mathcal{D}}_i^q \right)^{\frac{1}{1-q}}. \tag{8}$$

The factor $1/C$ regularizes the expression, keeping it finite in the $C \to \infty$ limit. This normalization corresponds to comparing the community's functional diversity with that of another community whose community-wide trait probability density function has the exact same support, but is uniform in that region.

In the main text, we exclusively rely on the inverse Simpson index for both species- and functional diversity—that is, $q = 2$. In the Supplementary Notes, the value of $q$ is always indicated. Additionally, see Supplementary Note 8 for alternative functional diversity metrics which also depend on the distance of trait values from one another in trait space.

**Analysis of the Galápagos land snail data**. The land snail data are organized in a table where each row corresponds to a single individual, and columns record individuals' species identity, morphological trait measurements, and the community they belong to. The latter is specified by the particular island the individual is from, and within the island, whether it came from an arid or humid vegetation zone. The distribution ranges of snail species never overlap across the two, so their species do not have the opportunity to interact with one another. This means that the species compositions of the humid and arid zones form effectively separate communities, and were treated here as such. Three small satellite islands were removed from the data (CH, ED, and GA). In addition, there were only two sampled individuals of the species *Naesiotus achatinellus*; those were also removed (all other species had at least 13 individuals sampled, with most having at least 20 specimens).

Snails were placed and their functional diversity evaluated in a two-dimensional trait space whose axes correspond to centroid size and the first PC axis of shell shape. This principal axis corresponds to whether shells are long and thin or compact and wide, and explains over 80% of shell shape variation. The species-level trait distribution functions were obtained either by fitting a binormal function using maximum likelihood estimates for the mean and the covariance matrix, or via kernel density estimation (Supplementary Note 7.1). These were then merged into a community-wise trait probability density function, whose functional diversity was evaluated as described above. Finally, corresponding species diversity values were normalized in each community by the number of potential local host plant species available for the snails.

**Reporting summary**. Further information on research design is available in the Nature Research Reporting Summary linked to this article.

## Data availability

The dataset of the Galápagos land snail communities can be accessed from: https://www.github.com/dysordys/phenotypediv.

## Code availability

All code for simulations and data analysis to replicate our results can be accessed from: https://www.github.com/dysordys/phenotypediv.

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

## Acknowledgements
We thank C. de Mazancourt, M. Loreau, A. Hendry, and G. Meszéna for discussions, and F. Barraquand for comments on an earlier manuscript version. Snail specimens were collected under permits from the Charles Darwin Foundation and the Galápagos National Park Directorate who also provided logistical help that made this work possible (CDF: #044-06, GNPD: #PC-45-14, PC-52-15, PC-52-16). We thank C. Sevilla, W. Cabrera, N. Castillo, T. de Roy, N. Carter, C. Philip, C. Philson, Z. Root, Y. Roell, and B. Miller for the logistic, field, and laboratory assistance. GB was funded by the Swedish Research Council (grant VR 2017-05245). Galápagos fieldwork was supported by grants from the National Institute of General Medical Sciences of the National Institutes of Health (IDeA #P30 GM103324), the National Science Foundation (#1523540 to A.C.K. and #1751157 to C.E.P.), the National Geographic Society, the American Malacological Society, the Western Society of Malacology, the Conchologists of America, and the Systematics Research Fund to C.E.P. F.V.dP. was supported by a PhD fellowship from the Research Foundation Flanders. F.D.L. is supported by grants from the University of Namur (FSR Impulsionnel 48454E1); the Fund for Scientific Research, FNRS (PDR T.0048.16); and the ARC grant DIVERCE, a concerted research action from the special research fund (Convention 18/23-095).

## Author contributions
G.B. and F.D.L. conceived of the study, ran simulations, and analyzed data; G.B. developed the theory and wrote the manuscript and Supplementary Information; C.P. and A.K. provided snail data; G.B., C.P., A.K., F.V.P., and F.D.L. contributed to the final form of the manuscript and helped with data analysis.

## Funding

## Competing interests
The authors declare no competing interests.
