## [Peer Review File · Nature Communications]

The evolution of trait variance creates a tension between species diversity and functional diversityREVIEWER COMMENTS

Reviewer #1 (Remarks to the Author):

This is a very interesting paper in which a theoretical model is presented to explore the eco-evolutionary relationship between the number of species in a community and the functional diversity of the community. The model describes how species in more species-rich communities exhibit narrower niche breaths (in multidimensional trait space). I really enjoyed the conceptual framework and approach, and I think this paper adds a clear and important dimension to a highly active line of research. My questions/comments center largely around additional dimensions for consideration, and some questions regarding the utility of the snail example to empirically explore their theoretical model.

To what extent do the results from this species-level analysis corroborate or not some of the theoretical/empirical work that has been done on intraspecific scales? Bolnick et al. (2007; PNAS) comes to mind: in that study, niche expansion was associated with more generalized individuals (consistent with Van Valen's niche variation hypothesis). While the appeal of this paper is nonetheless strong, the tie back to related intraspecific concepts could be stronger.

How well does the land snail example support the broader conclusion that functional diversity should decrease with community species richness? Surely no positive relationship is found, but I'm not sure how compellingly the data "even suggest a negative one" (L 172). Species diversity per host species is very tight for most data points (Figure 4), with only two points (ES and SF, humid) with relatively more species per plant species. Is there enough variation in the number of species per plant species to compellingly explore this relationship?

Likewise, to what extent is shell variation consistent with functional variation? The authors explain that shell morphology is 'positively correlated with habitat heterogeneity and negatively correlated with the number of co-occurring congener species' (L 145-148). But, there isn't a compelling case that shell morphology correlates with microhabitat specialization (although this seems to be implied?). Likewise, they state that species segregate according to shell morphology (L150-152), and that such segregation shows that shell size and shape 'mediates competition' (L 152). But, how does this segregation in trait space support that? In the associated figure (Fig. S10), it's clear that species segregate in trait space, but isn't this just an expected property of individuals belonging to separate species? Phenotypic divergence is expected among different species; the more time since the MRCA, the more divergence we expect. Further down (paragraph starting at L 157), connection between morphology and microhabitat use is suggested based on expert communication, but data to this effect are not presented. Bolnick et al. (2007) make a compelling argument for why morphological variation can be a poor proxy for niche variation (in particular, it cannot account for behavior and complex phenotype-function interactions), so without more information, I find this connection in snails a bit tenuous.

Another question I had was whether the model assumes that niche dimensions are uncorrelated. The model shows how multidimensional functional diversity declines with the number of species to reduce competitive overlap, but does this consider whether niche dimensions can independently be modulated. If a species specializes on a dietary type, for example, to reduce competitive overlap with other species, might it require expansion in some other niche dimension (for example, spatial habitat use) to access that specific prey? To the extent that resources are shared (or not) across niche dimensions, how might the implications for functional diversity vary?

Minor Comments:

L 24: proxy variable, rather than intermediate variable? Or surrogate?

L 45: 'very' is unnecessary

L 50: Change e.g. to "plasticity, for example via behavioral changes..."

L 63: Change 'which' to 'that' (same in L 70)

L 83: Put the i.e., portion in parentheses or separate those portions as independent clauses after colons or semicolons.

L 243: add comma after e.g. (please note that there is inconsistency – add commas after i.e., and e.g., throughout).

Reviewer #2 (Remarks to the Author):

I read this manuscript with great interest. It explores the relationship between species diversity and functional diversity. This might at first sight be considered as a question of peripheral interest in community ecology. Indeed, and as stated in the manuscript, it may be trivially expected that both types of diversity should correlate positively. Using eco-evolutionary modelling, this study shows that this is not the case. This result, which is supported by empirical data, is not only important for biodiversity-ecosystem functioning research, but as well for the key question of species coexistence in communities. In general, I found the manuscript well written and the message clear despite the complexity of the underlying model. In this respect, the model given in the manuscript is sound, but it could be better explained for non-specialists (note that it was not possible for me to review the modelling part in the SI because of time constraints). Importantly, the inclusion of the empirical results is a key strength of this contribution. At the end, we have model predictions and empirical results that concur in showing that functional diversity does not trivially increase with diversity, a result important for our understanding of species coexistence and of ecosystem functioning.

I have however a basic problem with the manuscript in its current state: it is difficult to evaluate to what extent the congruent modelling results are general and not simply the outcome of “particular” choice of parameterization. Of course, this is a universal critic with modelling, but I think that this aspect should be better handled here: the assumptions of the model and the justification of the choice of the parameters should be better stated. This question arises when examining the Fig. 2: The relationship between equilibrium trait distribution and per capita growth rate is very clear in panel B but puzzling in panel A, with positive trait values in regions of negative r . It is explained in the legend that “trait breadths evolve as a balance between the trait-enhancing vs. trait-pruning influence of these processes”, but these processes and their “enhancing” and “pruning” influences remains obscure. The processes underlying the evolution of trait variance (a novel aspect of this work) has to be better elaborated. Currently, I find the explanations in lines 96-108 insufficient: is the shrinkage of positive growth in species rich communities a general result or the outcome of the particular way the model is parameterized? I have to say that I find the modelling part sensible and impressive, and the results interesting and important, but I miss a convincing argument/explanation for the strong decrease in the breadth of the trait distribution in panel B. You should be more explicit concerning your statement on line 109 that your results are robust to changes in model’s parameters and assumptions.

I have some minor comments on the manuscript:

L. 13: I guess: “is not just trivial”?

L. 46-47: I would provide references for this statement.

Fig. 1 panel D: Why do species have disconnected distributions?

Legend of Fig. 2 and 3: I would remove “after substituting in ...” and “Intrinsic growth rates are given by...”, respectively. This could be stated in the corresponding SI.

L. 188-189: This is a very interesting proposition that is linked to the results in Fig. S9. It should be possible – and interesting - to see how the density-diversity relationship evolves with time in your model.

Reviewer #3 (Remarks to the Author):

Comments to "The evolution of trait variance creates a tension between species diversity and functional diversity" by Barabás et al.

The authors challenge the view that species diversity promotes functional diversity and demonstrate using a theoretical model that it might often be more expected that functional diversity declines with species diversity, illustrating that expectations from species-poor versus species-rich communities may not necessarily match. The authors mention that this positive relationship might be nuanced due to (1) environmental variation and (2) evolution. The latter, because evolution encompasses changes in ITV that may also be driven by species diversity (species interactions). More species within a community result in more stronger (?) competition, and species try to avoid this by narrowing their niche resulting in lower functional diversity with higher species diversity. When competition is sufficiently strong, it may thus decrease ITV width resulting in disrupting positive relationship, and this is also demonstrated by the theoretical model. Although the authors mention that it is safer to say that often no relationship will be found. This may actually suggest that functional diversity is the same for species-poor and species-rich communities. With the difference that in species-rich communities, it is species diversity that fills up the available niches, while in species-poor communities it is genetic diversity filling up the available niches.

Overall, the authors address an interesting question, the paper is well written (but quite dense). I have a few larger comments and some minor ones.

1. I was wondering if competition is not modelled too strong? And whether there is enough difference between competition widths of only 0.05 units apart? Does 0.1 reflect such low competition if 0.15 reflects high competition? I wonder if the narrow trait distributions are perhaps the result of very strong competition, and if this thus reflects the negative relationship? How do the trait distributions look like of these two competition treatments? For example, will the 6-species community illustrated in Fig. 2B have slightly larger trait widths?

2. I was missing a 'control' scenario in the study, where the results (negative relationship) found are contrasted against a scenario where no competition is present. Should then a positive relationship be observed? The authors mention that a negative relationship is found because species evolve narrower niches due to competition, but I wonder if a positive relationship would be found when no competition is present?

3. In another study by the authors (Barabás et al. 2016), they showed that species differing in the width of their trait distribution could also reduce competition. I somehow was expecting to also find such outcome in the current study, especially if species can evolve their trait width. Why would this not have been an optimal strategy?

Minor comments:

In the abstract the authors mention this very specific example of the cat family Felidae, but then do not repeat this or explain a bit more in detail in the introduction, which I think is a bit odd.

L60-62: For me it was not clear how the communities could vary in species diversity but have near-identical biotic environmental conditions. Because different species diversity would also result in different biotic environmental conditions. Perhaps the authors can clarify what they mean with these biotic environmental conditions?

For the same sentence, why are independent evolutionary histories needed? Later the authors refer to this as 'evolved independently' (L138). It is meant that there is no gene flow occurring between the populations? But in principle they could have evolved 'independently' from a common ancestor, so sharing the same evolutionary history?

L183-186: This is actually interesting, that while a negative relationship is found between species and functional diversity, there is still a positive relationship between some ecosystem functions and

species diversity. Does this mean that for these functions, there was also a positive relationship between functional diversity and resource use and biomass production? This shows that more species diversity can still be important for ecosystem functioning, and we should still try to maintain biodiversity. I think this would be important to mention a bit more clearly in the discussion, because somehow in a first read I interpreted the conclusion as if species diversity doesn't result in more functional diversity, than why should we preserve species diversity? I don't think that is the message we should convey.

For the empirical example, I was wondering if the relationship differed between the two habitat types in the empirical study presented? And thus if these relationships could not only be scale-dependent, but perhaps also habitat-dependent (on the harshness of the environment)?

L192-193: Perhaps a bit more information could be added what the authors mean with this scale-dependency of biodiversity on function. Which scales are they referring to? Is it reflection spatial scales linked to the model system used? Or is it more on how functional and species diversity scales to ecosystem functioning?

Fig 1D: I am not sure if I understand this figure. Is this reflecting that the 6 species have multimodal trait distributions? But in the main text it is mentioned that trait distributions are normal (L124). But then why do most species have these distinct trait values in trait space? Or is this not reflecting a 6-species community?

Fig 2B: why is fitness so low between the first and second species and between the fifth and sixth species, but not between the other species? I also was wondering why the species did not fulfil the full trait space, but keeping distinct trait distributions? What makes this 6-species scenario so different from the 3-species that they evolve such narrow trait distributions? Is this related to the value of the competition width? In addition, all 6 species seem to have similar niche width, but this was not the case for the 3-species scenario. Why did the species evolve similar niche width?

Fig4: In the caption, the authors mention that the pattern is not expected to be linear? Meaning a logarithm relationship is expected in the first place (from previous studies). Maybe this could be emphasized in the introduction also?

Response to reviewer comments

Manuscript number: NCOMMS-21-26802

“The evolution of trait variance creates a tension between species diversity and functional diversity”
György Barabás, Christine Parent, Andrew Kraemer, Frederik van de Perre & Frederik De Laender

Reviewer 1

This is a very interesting paper in which a theoretical model is presented to explore the eco-evolutionary relationship between the number of species in a community and the functional diversity of the community. The model describes how species in more species-rich communities exhibit narrower niche breadths (in multidimensional trait space). I really enjoyed the conceptual framework and approach, and I think this paper adds a clear and important dimension to a highly active line of research. My questions/comments center largely around additional dimensions for consideration, and some questions regarding the utility of the snail example to empirically explore their theoretical model.

We thank the Reviewer for the encouraging comments and the constructive questions about our work and model system. These have made us think more carefully about connecting our results with the literature, as well as to explain and justify better why the shell morphology of the Galápagos land snails is an appropriate functional trait in the context of the study. Please find our detailed responses below.

To what extent do the results from this species-level analysis corroborate or not some of the theoretical/empirical work that has been done on intraspecific scales? Bolnick et al. (2007; PNAS) comes to mind: in that study, niche expansion was associated with more generalized individuals (consistent with Van Valen’s niche variation hypothesis). While the appeal of this paper is nonetheless strong, the tie back to related intraspecific concepts could be stronger.

This is an important suggestion. In the original manuscript, we have made just one single allusion to results on intraspecific variation: Siefert et al. (2015) have shown that there is a negative relationship between the number of species in a community and the intraspecific trait breadths of those species. There is clearly room for a broader discussion of how our results connect to intraspecific concepts.

Bolnick et al. (2007), cited by the Reviewer, make two interesting claims that are relevant for our study. First, intraspecific trait breadths do expand after experiencing ecological release (where “traits” are meant in the broad sense, encompassing both morphological and behavioral characteristics). This is consistent with Siefert et al. (2015), because the reason for the inverse relationship between species richness and intraspecific variability might very well come from species expanding in their trait breadths when competitors are absent.

Second, Bolnick et al. (2007) observe that between-individual variability in diet increases with total population-level niche breadth. While our model might be too abstract to read much into any connection with this observation, this is in fact how the model works. In the model, individuals have one given trait (represented by a single point in the multidimensional trait space), and their diet breadths around that trait are given by the matrix W (See the Methods in the main text). The total, population-level diet breadth arises as the sum of the diet breadths of its individuals. That is, a more generalist population achieves its generality by harboring individuals that are more heterogeneous in their resource use, and not because its individuals are themselves better generalists. We now discuss these points in our revision (lines 280-291).

How well does the land snail example support the broader conclusion that functional diversity should decrease with community species richness? Surely no positive relationship is found, but I'm not sure how compellingly the data "even suggest a negative one" (L 172). Species diversity per host species is very tight for most data points (Figure 4), with only two points (ES and SF, humid) with relatively more species per plant species. Is there enough variation in the number of species per plant species to compellingly explore this relationship?

We have certainly not been very careful with discussing the subtleties of our claim. To remediate, we now say that there is no support for a positive relationship. Then, in the Discussion, we mention three lines of evidence providing some support for the relationship being actually negative.

To summarize our arguments: first, log-transforming the data along the x -axis still results in an overall negative slope. This increases the variation in the x -direction and also brings its distribution closer to normal. (A Shapiro-Wilk test fails to reject the null hypothesis that the log-transformed data are from a normal distribution; by contrast, this is easily rejected on the original data. See the SI, Section 7.2 and Figure S15.)

Second, removing any or both of the two "outliers" in the data with outstandingly high species diversity and outstandingly low functional diversity (ES humid and SF humid) does not suddenly result in a significant positive slope. In fact, removing only ES humid still gives a negative slope, while removing only SF humid yields a slope of zero. Only when removing both at the same time do we get a positive slope, but the result is nonsignificant. This suggests that the observed pattern is not being excessively driven by those two points.

Third, since the number of samples per species in our data do not reflect their relative abundances across the islands, we had to perform a randomization test, whereby we randomly assign abundances to the species and repeat the regression (SI, Section 7.2). This turned out to produce a negative slope more than 91% of the time. That is, the negative slope is very robust to altering the underlying data.

None of these observations force the interpretation that there is an overall negative relationship between species- and functional diversity in the snail data. Yet, when taken together, they do at least increase the plausibility of such a claim. We have expanded our text to explain these points in detail (lines 311-325).

Likewise, to what extent is shell variation consistent with functional variation? The authors explain that shell morphology is 'positively correlated with habitat heterogeneity and negatively correlated with the number of co-occurring congener species' (L 145-148). But, there isn't a compelling case that shell morphology correlates with microhabitat specialization (although this seems to be implied?).

As guessed by the Reviewer, there is indeed clear evidence of habitat specialization—which we have neglected to discuss and cite. In a nutshell, snails with long, slender shells live in dry habitats; snails with bulky and round shells live in humid habitats; and there are also intermediate varieties which live in semi-humid environments (Parent 2008, Kraemer et al. 2021). Additionally, though more peripherally for our purposes, there is evidence of a tight correlation between shell coloration and local background coloration (Kraemer et al. 2019). We have added the evidence that shell morphology correlates with microhabitat specialization to the main text (lines 194-204 and 292-310).

Likewise, they state that species segregate according to shell morphology (L150-152), and that such segregation shows that shell size and shape 'mediates competition' (L 152). But, how does this segregation in trait space support that? In the associated figure (Fig. S10), it's clear that species segregate in trait space, but isn't this just an expected property of individuals belonging to separate species? Phenotypic divergence is expected among different species; the more time since the MRCA, the more divergence we expect.

Morphological divergence is indeed expected across independent lineages over time in the absence of selection. That said, there are good reasons to think that the observed trait segregation is due to selection. There is the evidence mentioned above about shell morphology being an important indicator of microhabitat specialization: Galápagos land snail shell shape is associated with elevational variation in moisture levels, so that species with more slender shells (and a smaller shell opening relative to shell size) tend to be found at lower elevations, whereas species that are more conical in shape tend to be found at higher elevations where humidity levels are higher (Parent 2008). If that is the case, then snails are potentially competing for similar habitats if their shell morphologies are similar (lines 194-204).

The data can also be used to at least heuristically support the idea that the observed divergence is due to character displacement instead of neutral evolution. Namely, character divergence is only evident if one looks in the two-dimensional trait space generated by the shell centroid size and the first principal component of shell shape (Figure S12). Looking at only one of these traits at a time reveals no phenotypic divergence. Indeed, when we initially embarked on this project, we only looked at one trait at a time (either shell size or shell shape), and could not see any evidence of character differentiation. This questioned the whole idea that shell morphology is important for microhabitat competition—until we started looking in the two-dimensional trait space spanned by both shell size and shell shape.

Further down (paragraph starting at L 157), connection between morphology and microhabitat use is suggested based on expert communication, but data to this effect are not presented. Bolnick et al. (2007) make a compelling argument for why morphological variation can be a poor proxy for niche variation (in particular, it cannot account for behavior and complex phenotype-function interactions), so without more information, I find this connection in snails a bit tenuous.

We thank the Reviewer for mentioning this, because the general point made by Bolnick et al. (2007) is indeed very important to keep in mind whenever we try to estimate niche variation from purely morphological data. In the case of the Galápagos land snails, there are several particular circumstances which increase the credibility of shell morphology being indicative of ecological roles. The following lines of evidence support this statement (added to our text at lines 292-310):

- The concept of “behavior” applies less to land snails than to other invertebrates and to vertebrates. This is because these snails stay inactive for prolonged periods of time, attached to surfaces for months at a time (until it rains).
- The morphological shell traits we consider (and have been typically considered for this model system), happen to be especially good indicators of the snails’ selective environments. Snail shells serve numerous functions: they provide protection against predation (Goodfriend 1986) and water loss (Machin 1967, Goodfriend 1986), and they facilitate heat dissipation (McMahon 1990, Rees and Hand 1990). These advantages must be balanced against the metabolic costs of building, maintaining, and transporting the shell through a dynamic environment. In particular, shell size and shape are strongly selected by local environmental conditions for many mollusks (e.g., Newkirk and Doyle 1975, Seeley 1986).
- The ecological role played by shell morphology is also implied from the cost involved in building elongated shells. Elongated shells are more costly to produce for the snails than rounder shells because of their higher surface area to volume ratio (Goodfriend 1986). Thus, if building cost formed the sole selective pressure, snails would be predicted to all have rounder shells. However, elongated shells present important advantages such as decreased water loss and increased heat dissipation due to their smaller aperture and greater surface-to-volume ratio, respectively (Schmidt-Nielsen et al. 1971). Thus, the very presence of elongated shells implies they must confer an advantage over rounder shapes.

Another question I had was whether the model assumes that niche dimensions are uncorrelated. The model shows how multidimensional functional diversity declines with the number of species to reduce competitive overlap, but does this consider whether niche dimensions can independently be modulated. If a species specializes on a dietary type, for example, to reduce competitive overlap with other species, might it require expansion in some other niche dimension (for example, spatial habitat use) to access that specific prey? To the extent that resources are shared (or not) across niche dimensions, how might the implications for functional diversity vary?

By default, the model is agnostic about any correlations between the niche dimensions. This means that, while no genetic correlation structure is imposed by hand, it may still emerge in response to the selective pressure exerted by, e.g., other species in the community. An example is provided by Figure 1C in the main text, which illustrates a model simulation outcome. There is nothing in the model that would prescribe the species to adopt one genetic covariance structure or another. Yet, the three surviving species each assume a visibly nonzero covariance. (Zero covariance would correspond to the 95% contour line being a circle instead of an ellipse.) This is because the species would want to evolve towards the middle where intrinsic growth rates are highest, but are prevented from doing so by the others. In this case, the best compromise between achieving high growth rates without experiencing too much competition leads to the emergence of a nonzero genetic covariance structure for each species.

Minor Comments:

L 24: proxy variable, rather than intermediate variable? Or surrogate?

We have changed it to “surrogate” (line 23).

L 45: 'very' is unnecessary

We have removed it (line 44).

L 50: Change e.g. to “plasticity, for example via behavioral changes. . .”

Changed to “While trait narrowing can be due to plasticity, for example via behavioral changes or changes of resource preference ...” (lines 49-50).

L 63: Change 'which' to 'that' (same in L 70)

Changed, in both places (lines 63 and 70).

L 83: Put the i.e., portion in parentheses or separate those portions as independent clauses after colons or semicolons.

We have put the clause in parentheses (line 79).

L 243: add comma after e.g. (please note that there is inconsistency – add commas after i.e., and e.g., throughout).

We have changed it, and have gone through all other instances to make sure about the commas (this was done both in the main text and the Supplement).

Reviewer 2

I read this manuscript with great interest. It explores the relationship between species diversity and functional diversity. This might at first sight be considered as a question of peripheral interest in community ecology. Indeed, and as stated in the manuscript, it may be trivially expected that both types of diversity should correlate positively. Using eco-evolutionary modelling, this study shows that this is not the case. This result, which is supported by empirical data, is not only important for biodiversity-ecosystem functioning research, but as well for the key question of species coexistence in communities. In general, I found the manuscript well written and the message clear despite the complexity of the underlying model. In this respect, the model given in the manuscript is sound, but it could be better explained for non-specialists (note that it was not possible for me to review the modelling part in the SI because of time constraints). Importantly, the inclusion of the empirical results is a key strength of this contribution. At the end, we have model predictions and empirical results that concur in showing that functional diversity does not trivially increase with diversity, a result important for our understanding of species coexistence and of ecosystem functioning.

We thank the Reviewer for the positive comments, as well as the constructive critical remarks. The model can indeed look intimidating, and it has many moving parts to keep track of. Below we address in more detail how we tried to elucidate the model in our text, and how we expanded our exploration of its parameter space. But even before that, one thing we have done is to generate videos of the dynamics, to give a better intuitive feel for how they unfold. Our manuscript only contained snapshots of communities at steady state (Figure 1), without showing how those steady states actually arise from initial conditions. We hope that these videos will help convey how the model works. They can be viewed at the Github repository of this project: <https://www.github.com/dysordys/phenotypediv>.

I have however a basic problem with the manuscript in its current state: it is difficult to evaluate to what extent the congruent modelling results are general and not simply the outcome of “particular” choice of parameterization. Of course, this is a universal critic with modelling, but I think that this aspect should be better handled here: the assumptions of the model and the justification of the choice of the parameters should be better stated.

This is a very understandable and legitimate concern—especially in connection with our model, which has a large array of ingredient functions and parameter settings which can be altered. We have already tried to address some of these concerns, by systematically varying the number of trait dimensions, the initial number of species, the initial genetic covariance matrices, the environmental covariance matrices, the competition width, and the shape of the intrinsic growth function in a fully factorial combination. These explorations revealed that our results are fairly robust (Figure 3). However, there are still assumptions left in the model that were never relaxed. For instance, the degree of competition between two phenotypes was always assumed to be a Gaussian function of the distance in phenotype space (SI, Eq. 72). This assumption was made because it allows one to analytically evaluate the integrals in the model’s equations.

Following the Reviewer’s advice, we decided to try and test the limits of the model, in terms of overturning the negative relationship between species- and functional diversity. To this end, we implemented two new model variants. First, we replaced the Gaussian competition kernel with an asymmetric one. It is described by a scaled and shifted Gamma distribution, and it does not achieve its maximum at zero trait difference. Second, we abandoned the idea of localized competition (where phenotypes which are too far from one another do not interact appreciably) and implemented a hierarchical competition-mortality tradeoff model (Adler and Mosquera 2000). The two competition kernels are depicted below, in Figure R1.

Figure R1: Alternative forms for the competition kernel $a(z, z')$. Left: $a(z, z') = 10\Gamma((z - z')/\omega + 10, 4, 1/2)$, an asymmetric function where $\Gamma(z, \alpha, \beta)$ is the probability density function of the gamma distribution with shape parameter α and rate parameter β , and ω controls the width of the function (here, $\omega = 0.01$). Right: the hierarchical competition function $a(z, z') = (\tanh((z - z')/\omega) + 1)/2$ with a width of $\omega = 5$ (controlling how quickly the function transitions from low to high values). In this case, two phenotypes with substantially different trait values experience competition very differently: the phenotype with the lower trait value experiences no competition, whereas the one with the higher trait experiences strong competition. Biologically, this might correspond to competition for light between individuals of different heights, with height increasing in the negative direction along the trait axis.

Since these functions cannot be analytically jointly integrated with the trait distributions (SI, Section 4), one must resort to numerical integration to evaluate the right-hand sides of Eqs. 59-61 (also written out at the top of the Methods). This effectively restricts the analysis to one-dimensional trait spaces, because the computational effort increases exponentially with the number of dimensions over which one integrates. But for that case, we have modified and ran our model. The results are in Figure R2.

As seen, having introduced some asymmetry into the competition and having made it peak away from zero trait difference did not generate any qualitative differences from Figure 3 in the main text: there is still an overall negative relationship between species- and functional diversity. However, the relationship flips around and is positive for the case of hierarchical competition. This indicates the possible limits to our results, which may only hold under localized competition in trait space.

The reason for the inverted diversity pattern is also interesting, and is illustrated in Figure R3. The lowest species in the hierarchy (i.e., the one with the largest mean trait value) always evolves a very broad trait distribution, regardless of its number of competitors. This means that trait breadths do not universally shrink in response to interspecific competition, resulting in a positive relationship between species- and functional diversity. In fact, there is a more general positive relationship between trait mean and trait breadth: species that are on average lower in the hierarchy (higher trait values) have larger trait variance.

We have added these new results to the revised Supplement (Section 6.4), and are now pointing out that our results may shift in communities where competition materializes differently from the commonly used localized forms (represented by either the Gaussian or Gamma kernels in our case). We do so in the Results in lines 132-152. In the Discussion, we discuss these findings further and also propose, as an avenue for future research, to expand our analyses to multiple kinds of interaction kernels (lines 229-239).

Figure R2: Functional diversity against species diversity, like in Figure 3 of the main text but for the two competition kernel forms of Figure R1 (rows). Colors represent smaller (yellow) and larger (blue) values of the competition width ω . While the asymmetric case produces much the same pattern as before, the hierarchical model does not: in that case, functional diversity actually increases with species diversity. The intrinsic growth functions were either the quadratic $b(z) = 1 - 4z^2$ (asymmetric competition), or the monotonic $b(z) = 1 - \exp(z - 1/2)$ (hierarchical competition). This last choice means that the worse competitor a phenotype is in the hierarchy, the higher its growth rate, as per the standard assumptions of competition-mortality tradeoff models. However, there are diminishing returns beyond a point, because the function asymptotes at 1 for large values of z .

This question arises when examining the Fig. 2: The relationship between equilibrium trait distribution and per capita growth rate is very clear in panel B but puzzling in panel A, with positive trait values in regions of negative r . It is explained in the legend that “trait breadths evolve as a balance between the trait-enhancing vs. trait-pruning influence of these processes”, but these processes and their “enhancing” and “pruning” influences remains obscure.

We have improved on the explanation of this point—in the caption of Figure 2 as well, but mostly in the main text (lines 88-114). To understand what determines the equilibrium trait widths, let us take the leftmost species of Figure 2A as an example. Within this species, individuals with trait values close to the species-level mean around -0.25 are selected against (have negative growth rate); individuals with either larger or slightly smaller trait values are selected for (local maxima of the orange fitness curve), and individuals even more different from the species mean are again selected against. Intuitively, this should result in the trait distribution becoming bimodal with time. But it does not, because of the assumed random mating which restores the normal shape of the distribution even though the mean trait value is suboptimal. Instead, the trait variance increases. Yet, this increase does not continue indefinitely. That is because individuals with sufficiently extreme trait values have negative fitness: for very large trait

Figure R3: Final steady state of a 3-species (left) and a 8-species (right) community, under hierarchical competition $a(z, z') = (\tanh((z - z')/5) + 1)/2$ and intrinsic growth $b(z) = 1 - \exp(z - 1/2)$ (red dashed curve). Species' trait distributions are shown by the filled curves, truncated at 99.9% of the total density; the area under the curve is the total population abundance. Unlike in the case of localized competition in Figure 1 of the main text, only those species evolve narrow trait breadths whose mean trait values are low (meaning they are high up in the competitive hierarchy). The larger the trait means, the larger the trait variances get, which means that the original pattern of reduced functional diversity with increasing species diversity is overturned (see Figure R2).

variances, the negative selection against extreme trait values will counter-push to reduce the trait variance. The balance of the two processes then determines the trait variance achieved at equilibrium.

The processes underlying the evolution of trait variance (a novel aspect of this work) has to be better elaborated. Currently, I find the explanations in lines 96-108 insufficient: is the shrinkage of positive growth in species rich communities a general result or the outcome of the particular way the model is parameterized? I have to say that I find the modelling part sensible and impressive, and the results interesting and important, but I miss a convincing argument/explanation for the strong decrease in the breadth of the trait distribution in panel B. You should be more explicit concerning your statement on line 109 that your results are robust to changes in model's parameters and assumptions.

The shrinkage of trait variances in tightly packed communities is a general result that holds across all parameterizations (except when we replace localized competition with hierarchical competition; Figures R2-R3). This point is related to the previous one, and has caused confusion because of our overly terse treatment of how the model works in the main text. In response, we have now expanded our explanation of how trait (co-)variances evolve in the model, as explained above (lines 88-114). We also hope that the video animations will help in conveying of how this evolution works out dynamically.

There is one point to keep in mind about the shrinkage of the trait variance: the total variance P_i of species i is the sum of its genetic and environmental variances, so $P_i = G_i + E_i$. But since only the genetic component of the variance can evolve, and the lowest value it can achieve is zero, the total phenotypic variance will then still be equal to $P_i = E_i$. This is not necessarily small, in case the environmental variance component is large. That is why, both in Figure 3 in the main text and along the columns of Figure R2 above, we see that functional diversity drops less with species diversity in the presence of higher environmental variances (lines 124-131 in the revised main text). For illustration, we have chosen to depict a scenario in Figure 2 where the E_i are small. In case the E_i are larger, the final trait breadths

would also be larger, despite the fact that the fitness landscape would still look much the same as in Figure 2B.

I have some minor comments on the manuscript:

L. 13: I guess: “is not just trivial”?

It was supposed to have read “nontrivial”. What we meant is that we already know, from prior literature, that the relationship between species- and functional diversity can be more complex than one might think. What we argue here is that even more is true: the relationship is not just complex (“nontrivial”), but can be negative, which is not something that has been considered before.

Admittedly, the sentence was too complicated. In the revision, we decided to remove the extra clause altogether and simply write “The finding that the relationship between species- and functional diversity can be negative implies that...” (lines 12-14).

L. 46-47: I would provide references for this statement.

Our sentence was misleading—it read as if we were quoting a known statement from the literature. Instead, we are proposing the described mechanism (that greater species diversity may imply that individuals must avoid overly strong interactions with individuals from other species in order to persist), and then mention some evidence in favor of it. We have now clarified our text (lines 45-48).

Fig. 1 panel D: Why do species have disconnected distributions?

They do not—the problem is that there aren’t enough colors on a colorblind-friendly palette to represent all species with a different one. This is indeed confusing, and there are two possible solutions. First, we could state in the caption that repeated colors across different species are an artifact of not having enough colorblind-friendly colors. Second, we could stop using different colors for different species altogether, since their distributions can be clearly distinguished even in the absence of color-coding. For now, this is the choice we have decided to go with, and updated both Figures 1 and 2 (and their captions) accordingly.

Legend of Fig. 2 and 3: I would remove “after substituting in ...” and “Intrinsic growth rates are given by...”, respectively. This could be stated in the corresponding SI.

Yes, these are indeed more distracting than useful, especially that we would get the same qualitative outcome even if we used the other intrinsic growth function. We have removed them, and added this information to the Methods (lines 367-368; see also lines 388-391 for a similar statement on measuring diversity in the main text).

L. 188-189: This is a very interesting proposition that is linked to the results in Fig. S9. It should be possible – and interesting – to see how the density-diversity relationship evolves with time in your model.

We were very excited about this suggestion, and were expecting the time-dependence of the density-diversity relationship to reveal something quite interesting and relevant. Unfortunately, the results appear to be not so exciting after all. We are supplying some examples here, in Figure R4, summarizing the time evolution of the density-diversity relationship for communities of different initial species richness. Each community has been replicated 5 times, differing only in initial conditions. There are two discernible patterns seen. First, total community density settles more quickly than species diversity. That is, the community quickly reaches a steady total biomass, at which point it is only the relative densities which change around, affecting species diversity through changes in evenness. Second, within replicates of a given initial species richness, the density-diversity ratio eventually converges to the same value. This,

however, is not very surprising, reflecting only that the number and relative abundances of the eventually surviving species is largely determined by initial species richness.

Beyond that, it seems difficult to say anything of relevance about the observed pattern. In light of this, we have decided not to include these results in the manuscript. Nevertheless, if the Reviewer still feels there might be something interesting here, then we will be happy to include them in a revision.

Figure R4: Time evolution (x-axis) of the total community density (top row), species diversity as measured by the inverse Simpson index (middle row), and their ratio (bottom row). The time evolution of these quantities is mapped out for three different community types, each with a different number of initial species S (columns). Within each community type, there are five replicates (colors) which only differ from the others in initial conditions. The time scales are different for the three columns because species-rich communities take longer to reach steady state. Other parameters: number of trait dimensions $L = 1$, competition width $\omega = 0.15$, environmental trait variances are sampled uniformly and independently for each species from the interval $[0.005, 0.008]$, and the intrinsic growth function is quadratic: $b(z) = 1 - 4z^2$.

Reviewer 3

The authors challenge the view that species diversity promotes functional diversity and demonstrate using a theoretical model that it might often be more expected that functional diversity declines with species diversity, illustrating that expectations from species-poor versus species-rich communities may not necessarily match. The authors mention that this positive relationship might be nuanced due to (1) environmental variation and (2) evolution. The latter, because evolution encompasses changes in ITV that may also be driven by species diversity (species interactions). More species within a community result in more stronger (?) competition, and species try to avoid this by narrowing their niche resulting in lower functional diversity with higher species diversity. When competition is sufficiently strong, it may thus decrease ITV width resulting in disrupting positive relationship, and this is also demonstrated by the theoretical model. Although the authors mention that it is safer to say that often no relationship will be found. This may actually suggest that functional diversity is the same for species-poor and species-rich communities. With the difference that in species-rich communities, it is species diversity that fills up the available niches, while in species-poor communities it is genetic diversity filling up the available niches. Overall, the authors address an interesting question, the paper is well written (but quite dense). I have a few larger comments and some minor ones.

We thank the Reviewer for the excellent summary of our results, and the constructive and helpful comments. Please see our point-by-point responses below.

1. I was wondering if competition is not modelled too strong? And whether there is enough difference between competition widths of only 0.05 units apart? Does 0.1 reflect such low competition if 0.15 reflects high competition? I wonder if the narrow trait distributions are perhaps the result of very strong competition, and if this thus reflects the negative relationship? How do the trait distributions look like of these two competition treatments? For example, will the 6-species community illustrated in Fig. 2B have slightly larger trait widths?

There are two aspects to competition in our model as described by Eq. 72 in the SI. One is the *strength* it achieves, which is measured by the height of the function. In our formulation, we always scaled the function so that competition strength at zero trait difference is one: $a(z, z) = 1$. Changing this would not have affected diversity in any way, because stronger competition simply acts as a scaling factor to the equilibrium population densities with no effect on their relative values. In a Lotka–Volterra model for instance (which our ecological model ultimately falls under), the population densities N_i are governed by

$$\frac{dN_i}{dt} = N_i \left(b_i - \eta \sum_j \alpha_{ij} N_j \right),$$

where η is a generic dimensionless scaling factor setting the overall degree of competition by multiplying the original competition coefficients α_{ij} . The equilibrium densities are obtained by setting dN_i/dt to zero and solving for the N_i :

$$N_i^* = \frac{1}{\eta} \sum_j (\alpha^{-1})_{ij} b_j.$$

The equilibrium densities all get scaled by $1/\eta$, without any effect on the relative abundances. The effect of η therefore cancels from diversity metrics, since those depend on relative frequencies only.

The second aspect of competition is how *localized* it is: how far must two phenotypes be from each other in trait space for competition to get substantially reduced between them? This is controlled by the competition width ω , and is what we had been varying. If this width is small, then even a modest trait difference reduces competition to tolerable levels. If it is larger, then greater trait differences are needed. Ultimately, this means that more species can be packed into the trait space when ω is smaller.

Reducing ω has the same effect as increasing the size of the trait space. In our parameterization, θ measures the range of parameter space where positive growth is achievable in principle (i.e., without the burden of competition). We set this parameter to a constant $1/2$, which effectively restricts each trait dimension to the range $[-0.5, 0.5]$ —see the dashed curves in Figure 1 in the main text. But the effect of doubling the size of the trait space to range between $[-1, 1]$ is equivalent to keeping the trait space unchanged while reducing ω to half its value (Szabó and Meszéná 2006, Barabás and D’Andrea 2016).

Thus, choosing smaller or larger values of ω has the effect of altering the number of species that can be packed into the trait space. A lower ω allows more species, because they do not need to be as far apart from one another to avoid competition. This means higher potential species diversity values, which is visible in Figure 3 of the main text: lower values of ω lead to a larger maximum of species diversity.

To answer the Reviewer’s last question more directly: changing ω in Figure 2B would have had no effect on the observed intraspecific trait widths—as long as there would have been sufficiently many species to saturate the trait axis. We illustrate this point below, in Figure R5. The panels are completely identical in setup, except that the competition width increases from left to right. The only difference between the final equilibrium communities is how tightly they are packed. Otherwise, in terms of intraspecific trait widths or absolute population densities, they are very similar. Perhaps more importantly for our purposes, this also means that the negative relationship between species- and functional diversity is retained in all cases, though the extent of the decline does depend on ω (with larger competition widths yielding a steeper decline; see Figure 3 in the main text or the top row of Figure R2 here).

Figure R5: Communities with $\omega = 0.05$ (A), $\omega = 0.15$ (B), and $\omega = 0.25$ (C). Otherwise, all three scenarios are exactly identical, starting with $S = 30$ species and the same initial conditions under the quadratic intrinsic growth function $b(z) = 1 - 4z^2$ (dashed curves and right y-axes).

2. *I was missing a 'control' scenario in the study, where the results (negative relationship) found are contrasted against a scenario where no competition is present. Should then a positive relationship be observed? The authors mention that a negative relationship is found because species evolve narrower niches due to competition, but I wonder if a positive relationship would be found when no competition is present?*

Contrasting our results with those from a null model is a very valuable idea. There are multiple candidate null models one can compare our scenarios with. The first is the one suggested by the Reviewer, where no competition is present. Presumably, this means no *interspecific* competition: since something must ultimately bound the population densities of the species, there has to be some *intraspecific* competition, otherwise population densities would grow exponentially forever. Thus, we created a version of the model with only *intraspecific* competition, and with this competition being a constant $a_{ii} = 1$. Formally, this means that in Eqs. 86-88 of the SI, g_i , Q_i , β_{ij} , and Γ_{ij} are zero, and α_{ij} is also zero unless $i = j$. In

this null model, species keep their initial trait means and trait variances, and all they do is adjust their densities based on their self-interaction.

A second null model is to repeat our analyses of Figure 3 in the main text, with one change: trait variances cannot evolve ($Q_i = \Gamma_{ij} = 0$). The model then becomes identical to the one by Barabás and D’Andrea (2016), except that it is extended to multidimensional trait spaces. This null model isolates in particular the effect of trait variance evolution on the relationship between species- and functional diversity.

We have repeated our analyses using both null models (the code used to produce them is available at the Github repository of the project: <https://www.github.com/dysordys/phenotypediv>). In either case, functional diversity saturates with species diversity. This was expected: without the ability to reduce trait variance, adding more species to the community creates more trait space coverage. But beyond a point, the community is oversaturated with species and so there are diminishing returns for having even more of them.

After some deliberation, we have decided to only include results from the second null model in the main text (revised Figure 3 and caption; also lines 120-123). This is because the inclusion of the other, no-interaction null model made the graphs more messy without adding much beyond what was expected. For reference, we are showing here what the figure would look like with both null models shown (Figure R6). If the Reviewer thinks this version is better than the one in the main text without the first null model, then we will be happy to replace it.

3. In another study by the authors (Barabás and D’Andrea 2016), they showed that species differing in the width of their trait distribution could also reduce competition. I somehow was expecting to also find such outcome in the current study, especially if species can evolve their trait width. Why would this not have been an optimal strategy?

This is a very good observation—and, to be honest, we were expecting to see such patterns as well, at least in a couple of cases. Since we had tens of thousands of replicates, we were reasonably confident that such a scenario would have arisen at least a few times. So we sifted through our simulated dataset, looking for species pairs with very similar trait means but distinct trait variances. To our surprise, we have not found a single replicate with this configuration.

The fact is, the conditions for coexistence via such a generalist-specialist tradeoff are somewhat restrictive (Barabás and D’Andrea 2016), requiring the ratio of trait variances to be in a narrow window. The evolution of trait means makes this even more restrictive. It appears that further allowing for the evolution of trait variances introduces further restrictions—perhaps to the point of making such coexistence impossible. To test this, we have tried out some two-species runs in which the starting conditions favored (ecological) coexistence. While negative simulation results do not of course prove the impossibility of anything, we have not managed to find a single instance of coexistence at steady state. We therefore suspect that under trait variance evolution, the conditions for the spontaneous arising of the generalist-specialist tradeoff are at the very least so restrictive that one would usually not observe it emerging.

It may also be possible to rigorously prove its impossibility. However, such a proof might not be easy to obtain, because one must analyze a 6-dimensional dynamical system: two species, each with the three state variables of population density, trait mean, and trait variance. Speaking for ourselves, we have tried and failed to construct a theorem ruling out the evolution of generalist-specialist tradeoffs, so the possibility for this is still open. But based on the simulation results, such outcomes are likely quite rare.

We now discuss the possibility of generalist-specialist coexistence and its problems in lines 240-255.

Figure R6: As Figure 3 in the main text, but with the results from the no-interaction null model included. They are shown by purple lines (locally-weighted polynomial regression, with 99% confidence intervals). The results from the other, no-variance-evolution null model are shown by the colored lines (again via locally-weighted polynomial regression). They do not reach as far along the x-axis because without the trait breadths being allowed to shrink, only fewer species can be packed into the trait space, limiting species diversity.

Minor comments:

In the abstract the authors mention this very specific example of the cat family Felidae, but then do not repeat this or explain a bit more in detail in the introduction, which I think is a bit odd.

Good point—we have changed the example to something less specific (lines 2-4).

L60-62: For me it was not clear how the communities could vary in species diversity but have near-identical biotic environmental conditions. Because different species diversity would also result in different biotic environmental conditions. Perhaps the authors can clarify what they mean with these biotic environmental conditions?

The abiotic and biotic environment should indeed be very similar across the “replicates” (i.e., island-vegetation zone combinations). That is, environmental conditions (e.g., temperature, resource availability) but also the presence of other communities (vegetation types, birds) should be similar. The reason is that the same assumption underlies the theoretical model: for each color of Figure 3 for instance, we keep all

parameters constant *except* initial species richness. Implicitly, this means that the intrinsic growth curves, available resource types etc. are all unchanged. Therefore, to create a real-life analogue of such numerical experiments, one should find communities that are practically the same in terms of their environment *except* for the species diversity of the focal community. If two snail communities are much the same but one of them has an important predator species that is absent from the other, then we will not know whether any observed relationship between species- and functional diversity will not be due to the influence of the predator. By considering communities in near-identical environments, we have a better chance of controlling for such confounding factors. We have tried to clarify this on lines 56-65.

For the same sentence, why are independent evolutionary histories needed? Later the authors refer to this as 'evolved independently' (L138). It is meant that there is no gene flow occurring between the populations? But in principle they could have evolved 'independently' from a common ancestor, so sharing the same evolutionary history?

That is correct. The species may indeed have a common ancestor; what we need is a sufficiently long time without gene flow, otherwise we don't have proper, independent replicates. (Our model does not include dispersal, therefore the trait value of an individual only depends on the local dynamics). What we are after are the eco-evolutionary stable states, which require sufficiently long time that local dynamics have had their chance to fully play out. We have added a clarification to lines 61-65.

L183-186: This is actually interesting, that while a negative relationship is found between species and functional diversity, there is still a positive relationship between some ecosystem functions and species diversity. Does this mean that for these functions, there was also a positive relationship between functional diversity and resource use and biomass production? This shows that more species diversity can still be important for ecosystem functioning, and we should still try to maintain biodiversity. I think this would be important to mention a bit more clearly in the discussion, because somehow in a first read I interpreted the conclusion as if species diversity doesn't result in more functional diversity, then why should we preserve species diversity? I don't think that is the message we should convey.

We very much agree, and have decided to elevate the old Figure S9 to the main text (Figure 4), along with more explanation of this result (lines 153-166 and 256-265). Indeed, it is quite important to emphasize that the negative effect of species diversity on functional diversity does not imply reduced ecosystem functioning.

For the empirical example, I was wondering if the relationship differed between the two habitat types in the empirical study presented? And thus if these relationships could not only be scale-dependent, but perhaps also habitat-dependent (on the harshness of the environment)?

This is difficult to say for certain, due to the severe shortage of data points when disaggregating the data by habitat type. In terms of linear regression slopes (for whatever they are worth): the original data can be fitted with slope -11.4 . When disaggregating the data and repeating the regression only on the humid and arid regions, we get -4.85 (humid only) and 11.1 (arid only). On the face of it, this is quite a stark difference; and, in case we subscribe to the arid regions being more "harsh" environments, means that the relationship is flipped from negative to positive for harsher environments. However, looking at the data points in Figure 5 of the main text shows that no such conclusion can really be drawn from the data. At this stage therefore, we would rather avoid making claims about the effects of habitat type on our results.

L192-193: Perhaps a bit more information could be added what the authors mean with this scale-dependency of biodiversity on function. Which scales are they referring to? Is it reflection spatial scales linked to the model system used? Or is it more on how functional and species diversity scales to ecosystem functioning?

Our text here was confusing—by “scale” we meant time scale dependence. We have revised this, and expanded the section on the biodiversity-ecosystem functioning relationship, as advertised before (lines 256-265).

Fig 1D: I am not sure if I understand this figure. Is this reflecting that the 6 species have multimodal trait distributions? But in the main text it is mentioned that trait distributions are normal (L124). But then why do most species have these distinct trait values in trait space? Or is this not reflecting a 6-species community?

We completely neglected to clarify that in Figure 1D, we started out with $S = 25$ species initially, out of which 19 survived. Each distinct small ellipse represents the 95%-contour of the trait distribution of a different species. Furthermore, we probably also caused more confusion by re-using the same colors for different species (Reviewer 2 also pointed out the same problem). The reason is that there aren't sufficiently many colorblind-friendly colors in the palette we used (only 7 colors), which meant that we had to reuse them. In the revision, we simply represent all species with the same color, relying only on the contours of the different distribution curves to distinguish them.

Fig 2B: why is fitness so low between the first and second species and between the fifth and sixth species, but not between the other species? I also was wondering why the species did not fulfil the full trait space, but keeping distinct trait distributions? What makes this 6-species scenario so different from the 3-species that they evolve such narrow trait distributions? Is this related to the value of the competition width? In addition, all 6 species seem to have similar niche width, but this was not the case for the 3-species scenario. Why did the species evolve similar niche width?

First, the reason the overall fitness function dips so low near the edges is that the intrinsic growth rate function (dashed curve in Figure 1B) is nearly zero by that point. Subtracting the contribution of competition from this near-zero value means that the fitness function is very negative in those regions. Second, technically speaking, since species' trait distributions are normal, they always fill the entire trait axis. (On the figure we cut the tails of the distributions for visual aid; otherwise all species would overlap at the bottom.) And third, the 6 species evolve to be so narrow not because of the competition width *per se*, but because the trait space is saturated, in the sense that species are as close to one another as possible given the value of the competition width (Figure R5). This causes the overall fitness to be all negative, except at the mean trait values of the persisting species. This means that any trait values that are not right at the species mean are being selected against, leading to stabilizing selection which erodes the generic variances until they get to be zero. The reason this does not result in zero trait variances is that the total trait variance is the sum of the genetic and environmental variances—and the latter are not zero. But, since they are all very close in value (sampled uniformly from [0.005, 0.008]; see Methods), the trait widths are visually more or less equal in Figure 2B. By contrast, in Figure 2A the genetic variances are not zero and not equal to one another. In particular, the middle species experiences stronger disruptive selection than the two edge species, and therefore its intraspecific trait breadth also ends up being larger. Per request of Reviewer 2, who had a similar remark, we edited the the explanation of what causes trait variance to evolve in this way (lines 88-114).

Fig4: In the caption, the authors mention that the pattern is not expected to be linear? Meaning a logarithm relationship is expected in the first place (from previous studies). Maybe this could be emphasized in the introduction also?

We did not mean a log-relationship necessarily, but the predicted relationship from our model as summarized by Figure 3. We have now clarified this in the caption of Figure 5 (which used to be Figure 4 in the previous version).

References

- Adler, F. R., Mosquera, J., 2000. Is space necessary? Interference competition and limits to biodiversity. *Ecology* 81, 3226–3232.
- Barabás, G., D’Andrea, R., 2016. The effect of intraspecific variation and heritability on community pattern and robustness. *Ecology Letters* 19 (8), 977–986.
- Bolnick, D. I., Svanbäck, R., Araujo, M. S., Persson, L., 2007. Comparative support for the niche variation hypothesis that more generalized populations also are more heterogeneous. *Proceedings of the National Academy of Sciences USA* 104, 10075–10079.
- Goodfriend, G. A., 1986. Variation in Land-snail Shell form and Size and its Causes: a Review. *Systematic Biology* 35 (2), 204–223.
- Kraemer, A. C., Philip, C. W., Rankin, A. M., Parent, C. E., 2019. Trade-offs direct the evolution of coloration in Galápagos land snails. *Proceedings of the Royal Society B* 286, 20182278.
- Kraemer, A. C., Roell, Y. E., Shoobs, N. F., Parent, C. E., 2021. Does island ontogeny dictate both the accumulation of species richness and functional diversity? *Global Ecology and Biogeography*.
URL <https://doi.org/10.1111/geb.13420>
- Machin, J., 1967. Structural adaptation for reducing water-loss in three species of terrestrial snail. *Journal of Zoology* 152, 55–65.
- McMahon, R. F., 1990. Thermal tolerance, evaporative water loss, air-water oxygen consumption and zonation of intertidal prosobranchs: a new synthesis. In: *Progress in Littorinid and Muricid Biology*. Springer, Dordrecht, The Netherlands, pp. 241–260.
- Newkirk, G. F., Doyle, R. W., 1975. Genetic analysis of shell-shape variation in *Littorina saxatilis* on an environmental cline. *Marine Biology* 30, 227–237.
- Parent, C. E., 2008. Diversification on islands: bulimulid land snails of Galápagos. Ph.D. thesis, Simon Fraser University, Burnaby, Canada.
- Rees, B. B., Hand, S. C., 1990. Heat Dissipation, Gas Exchange and Acid-Base Status in the Land Snail *Oreohelix* During Short-Term Estivation. *Journal of Experimental Biology* 152, 77–92.
- Schmidt-Nielsen, K., Taylor, C. R., Shkolnik, A., 1971. Desert snails: problems of heat, water and food. *Journal of Experimental Biology* 55, 385–398.
- Seeley, R. H., 1986. Intense natural selection caused a rapid morphological transition in a living marine snail. *Proceedings of the National Academy of Sciences USA* 83, 6897–6901.
- Siefert, A., Violle, C., Chalmandrier, L., Albert, C. H., Taudiere, A., Fajardo, A., et al., 2015. A global meta-analysis of the relative extent of intraspecific trait variation in plant communities. *Ecology Letters* 18, 1406–1419.
- Szabó, P., Meszéna, G., 2006. Limiting similarity revisited. *Oikos* 112, 612–619.

REVIEWER COMMENTS

Reviewer #1 (Remarks to the Author):

I have read the revised manuscript and the response to reviewers. It's clear this is a much-improved manuscript. To distill some of my earlier concerns: There seemed to be a big organismal gap, with key details about the system missing, which made it challenging to really tie the system to the model. The authors did a very nice job rectifying this and I find the manuscript more compelling.

In my original review, I asked whether the model assumes that niche dimensions are uncorrelated. Specifically, the model shows how multidimensional functional diversity declines with the number of species, but the models this consider whether niche dimensions can independently be modulated. In their response, the authors said:

"By default, the model is agnostic about any correlations between the niche dimensions. This means that, while no genetic correlation structure is imposed by hand, it may still emerge in response to the selective pressure exerted by, e.g., other species in the community. An example is provided by Figure 1C in the main text, which illustrates a model simulation outcome. There is nothing in the model that would prescribe the species to adopt one genetic covariance structure or another. Yet, the three surviving species each assume a visibly nonzero covariance. (Zero covariance would correspond to the 95% contour line being a circle instead of an ellipse.) This is because the species would want to evolve towards the middle where intrinsic growth rates are highest, but are prevented from doing so by the others. In this case, the best compromise between achieving high growth rates without experiencing too much competition leads to the emergence of a nonzero genetic covariance structure for each species."

Space-permitting, this feels like an important point to add. Resources are for sure shared across niche dimensions, so I suspect I won't be the only reader to wonder about this.

L 186-188: I found the syntax and flow to be awkward in this sentence.

Reviewer #2 (Remarks to the Author):

I was pleasantly impressed by the quality and thoroughness of the answers to the reviewers. I read the revised manuscript and found that all my comments were adequately handled. I have the following minor comments:

Li. 105: I would remove "for instance" (redundant with "as an example").

Li. 114: I would be explicit about the "two processes" (e.g. "The balance of negative selection and of random mating then determines the trait variance achieved at equilibrium")

Fig. 4: In the legend, I would touch a word on the difference between quadratic and quartic intrinsic growth functions (refer to Method section (line 366) where I would introduce both terms). I suggest to present the results for only one growth function, and possibly for one width per dimension, as all results are qualitatively similar (in this case, it would of course not anymore be necessary to mention the difference in the growth functions).

Finally, there is one point that may be worth mentioning. A negative BEF relationship is not possible in the classical models of Tilman (PNAS 1997). The results in your Fig. 3 do not fit with this restriction. Since then, models have been proposed where a negative BEF is possible, notably in the work of Parain et al. (AmNat 2019). This may be worth mentioning.

Reviewer #3 (Remarks to the Author):

Comments to 'The evolution of trait variance creates a tension between species diversity and functional diversity'

This is the second time that I am reviewing this manuscript. I think the authors have done a very good job in addressing the comments of the reviewers, including mine. As mentioned previously, the authors address a very relevant research question and I think the results found are really interesting. The manuscript was already well-written, but I think it has improved even further. It was very easy to follow the reasoning the authors were making, which is very much appreciated. I really like the additions of the 'no evolution' model, the 'non-local competition' model and the 'asymmetric competition' model, and really interesting that the non-local competition resulted in finding a positive relationship. The authors mention that such type of competition is often found in herbaceous plants and trees. It kept me wondering if this is why many BEF studies find a positive relationship because most of them (at least to my knowledge) use a plant model system. This could be an interesting future research avenue to explore. Overall, I have no further comments left, I just found a few typo's that the authors might want to fix before publication.

L121: 'with no the evolution'  I think 'the' must be deleted here?

L215: the formulation 'need not promote' reads a bit odd  'does not promote'?

L313: word missing in 'that it a plausible hypothesis'  'that it is a plausible hypothesis'

Response to reviewer comments

Manuscript number: NCOMMS-21-26802A

“The evolution of trait variance creates a tension between species diversity and functional diversity”
György Barabás, Christine Parent, Andrew Kraemer, Frederik van de Perre & Frederik De Laender

Reviewer 1

I have read the revised manuscript and the response to reviewers. It's clear this is a much-improved manuscript. To distill some of my earlier concerns: There seemed to be a big organismal gap, with key details about the system missing, which made it challenging to really tie the system to the model. The authors did a very nice job rectifying this and I find the manuscript more compelling.

We thank the Reviewer for taking the time to once again thoroughly look through our work, and the positive evaluation. We are especially glad that the connection between the empirical system and the theoretical model feels more organic now!

In my original review, I asked whether the model assumes that niche dimensions are uncorrelated. Specifically, the model shows how multidimensional functional diversity declines with the number of species, but the models this consider whether niche dimensions can independently be modulated. In their response, the authors said:

“By default, the model is agnostic about any correlations between the niche dimensions. This means that, while no genetic correlation structure is imposed by hand, it may still emerge in response to the selective pressure exerted by, e.g., other species in the community. An example is provided by Figure 1C in the main text, which illustrates a model simulation outcome. There is nothing in the model that would prescribe the species to adopt one genetic covariance structure or another. Yet, the three surviving species each assume a visibly nonzero covariance. (Zero covariance would correspond to the 95% contour line being a circle instead of an ellipse.) This is because the species would want to evolve towards the middle where intrinsic growth rates are highest, but are prevented from doing so by the others. In this case, the best compromise between achieving high growth rates without experiencing too much competition leads to the emergence of a nonzero genetic covariance structure for each species.”

Space permitting, this feels like an important point to add. Resources are for sure shared across niche dimensions, so I suspect I won't be the only reader to wonder about this.

This is a good point—we have added the passage to our text (lines 116-125).

L 186-188: I found the syntax and flow to be awkward in this sentence.

We have broken up the sentence into two simpler sentences (lines 200-202): “Community age varies from 60 thousand to over three million years across the islands. Such long time spans are sufficient for substantial shell morphology evolution to have taken place.”

Reviewer 2

I was pleasantly impressed by the quality and thoroughness of the answers to the reviewers. I read the revised manuscript and found that all my comments were adequately handled.

We thank the Reviewer again for the attention dedicated to our work, and the positive evaluation. Please find our detailed responses below.

I have the following minor comments:

Li. 105: I would remove “for instance” (redundant with “as an example”).

We have removed it (line 106).

Li. 114: I would be explicit about the “two processes” (e.g. “The balance of negative selection and of random mating then determines the trait variance achieved at equilibrium”)

We modified the sentence as suggested (lines 114-115).

Fig. 4: In the legend, I would touch a word on the difference between quadratic and quartic intrinsic growth functions (refer to Method section (line 366) where I would introduce both terms).

I suggest to present the results for only one growth function, and possibly for one width per dimension, as all results are qualitatively similar (in this case, it would of course not anymore be necessary to mention the difference in the growth functions).

This is a very good idea—in fact, we now show only one width per dimension and only the results with a quadratic growth function in both Figures 3 and 4. We still show the full results in the Supplement; see the new Supplementary Figure 9. This way, we do not need to complicate matters in the main text by varying the intrinsic growth function at all.

Finally, there is one point that may be worth mentioning. A negative BEF relationship is not possible in the classical models of Tilman (PNAS 1997). The results in your Fig. 3 do not fit with this restriction. Since then, models have been proposed where a negative BEF is possible, notably in the work of Parain et al. (AmNat 2019). This may be worth mentioning.

We thank the Reviewer for bringing this article to our attention. While our Figure 3 does not show a BEF relationship, the general point remains valid, and we now cite Parain et al. (2019 Am Nat) when writing about our observed BEF relationship in the Discussion (lines 281-284).

Reviewer 3

This is the second time that I am reviewing this manuscript. I think the authors have done a very good job in addressing the comments of the reviewers, including mine. As mentioned previously, the authors address a very relevant research question and I think the results found are really interesting. The manuscript was already well-written, but I think it has improved even further. It was very easy to follow the reasoning the authors were making, which is very much appreciated. I really like the additions of the 'no evolution' model, the 'non-local competition' model and the 'asymmetric competition' model, and really interesting that the non-local competition resulted in finding a positive relationship.

We would like to thank the Reviewer again for the thorough engagement with our work, and the positive comments. Please see our point-by-point responses below.

The authors mention that such type of competition is often found in herbaceous plants and trees. It kept me wondering if this is why many BEF studies find a positive relationship because most of them (at least to my knowledge) use a plant model system. This could be an interesting future research avenue to explore.

This is a very interesting idea. Presumably, the Reviewer meant that the ecosystem function here is trait diversity itself (because other functions, such as total biomass or resource use, are still positively related to species diversity in our study). Then, what one would need to do is to check whether any of the existing species- vs. functional diversity observations done on trees and herbaceous plants happen to come from a comparable “natural experiment” as the land snail data we presented. The reason this is important is that without sufficient time for species to co-adapt their trait breadths to one another, one may easily find a positive relationship between species- and functional diversity (the key element in finding the negative relationship was the evolution of trait variances). It is certainly not out of the question that similar data to ours are available for taxa in which hierarchical competition is the norm. One could then see if, indeed, the relationship between species- and functional diversity *still* turns out positive, unlike in our work. We now mention this idea as a possibility in the Discussion (lines 251-253).

Overall, I have no further comments left, I just found a few typo's that the authors might want to fix before publication.

L121: 'with no the evolution' -> I think 'the' must be deleted here?

Correct; we have removed the extra 'the' (line 133).

L215: the formulation 'need not promote' reads a bit odd -> 'does not promote'?

We have rewritten the sentence saying “does not necessarily promote” (lines 228-229).

L313: word missing in 'that it a plausible hypothesis' -> 'that it is a plausible hypothesis'

Yes, the 'is' was missing. We now added it (line 332).